# Safety Verification of Decision-Tree Policies in Continuous Time

**Christian Schilling**
Aalborg University
Aalborg, Denmark
christianms@cs.aau.dk

**Anna Lukina**
TU Delft
Delft, The Netherlands
a.lukina@tudelft.nl

**Emir Demirović**
TU Delft
Delft, The Netherlands
e.demirovic@tudelft.nl

**Kim Guldstrand Larsen**
Aalborg University
Aalborg, Denmark
kgl@cs.aau.dk

## Abstract

Decision trees have gained popularity as interpretable surrogate models for learning-based control policies. However, providing safety guarantees for systems controlled by decision trees is an open challenge. We show that the problem is undecidable even for systems with the simplest dynamics, and **PSPACE**-complete for finite-horizon properties. The latter can be verified for discrete-time systems via bounded model checking. However, for continuous-time systems, such an approach requires discretization, thereby weakening the guarantees for the original system. This paper presents the first algorithm to directly verify decision-tree controlled systems in continuous time. The key aspect of our method is exploiting the decision-tree structure to propagate a set-based approximation through the decision nodes. We demonstrate the effectiveness of our approach by verifying safety of several decision trees distilled to imitate neural-network policies for nonlinear systems.

## 1 Introduction

Deep reinforcement learning has shown success in deriving control policies for *nonlinear* systems for which classical optimal control theory provides no solution [Mnih et al., 2015]. Despite impressive performance, there are two key drawbacks: (i) the resulting policy is difficult to interpret, and (ii) there are no guarantees that the system always reaches the desired goal and avoids unsafe states. The interpretability challenge has led to a review of *decision trees* as well-performing, compact, and transparent surrogates for deep neural networks [Ashok et al., 2020, 2019, David et al., 2015, Alamdari et al., 2020, Vos and Verwer, 2023]. To provide safety guarantees, prior work proposed to encode a discrete, time-bounded system controlled by a decision-tree policy as a set of logical constraints [Bastani et al., 2018]. However, this approach does not apply to continuous-time dynamics.

**Example 1.** *To illustrate the problem, we consider a quadrotor system [Ivanov et al., 2019] in Figure 1(a), where the task is to follow the brown reference trajectory from $(0,0)$ and stay within the blue dashed safety corridor. Illustrated schematically in Figure 1(b), we may observe a trajectory that exhibits safe behavior for each discrete time point but leaves the safety corridor in between. Since the trajectory returns quickly enough, this violation is missed by a discrete-time analysis.*

In this work, we address, for the first time, the problem of verifying safety (reach-avoid) properties of a decision-tree controlled system (DTCS) with (nonlinear) *continuous-time* dynamics. We propose an algorithm that exploits the structure of the decision-tree policy and propagates sets of reachable states,

37th Conference on Neural Information Processing Systems (NeurIPS 2023).

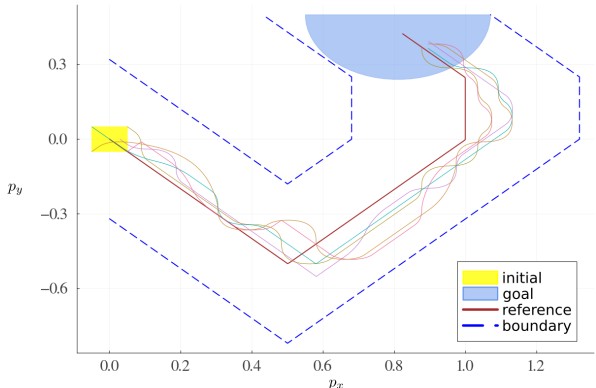

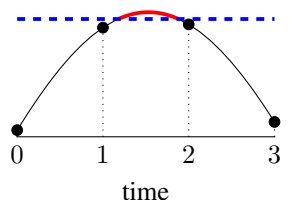

(a) Sketch of the quadrotor system ($x/y$ plane) with sample trajectories.

(b) Discrete-time analysis may miss property violations of continuous systems. The threshold (dashed blue line) is breached for a short time (red curve), which is not detected at the discrete time points (black dots).

Figure 1: A sketch of the quadrotor system and motivation for continuous-time analysis.

splitting at the decision boundaries. This way we obtain a sound enclosure of the true reachable states even for continuous dynamics, which guarantees safe goal reachability for all behaviors of the controlled system. Furthermore, our approach naturally generalizes to discrete-time systems, for which, compared to the aforementioned approach based on logical encoding [Bastani et al., 2018], our approach provides visual interpretability of the verification process.

To summarize our contributions, we first provide an algorithm to verify a decision-tree policy of a nonlinear continuous-time system. Our general algorithm is parameterized in two procedures for respectively analyzing the dynamical system and the policy. For these procedures, we formulate sufficient conditions for the algorithm to be sound and relatively complete. Second, we describe an instantiation of the algorithm for nonlinear systems based on Taylor models [Berz and Makino, 1998]. In a nutshell, this procedure propagates a set of states through the system dynamics. For analyzing the decision-tree policy, we focus on the common class of axis-aligned predicates ("$x \leq c$") and propose an instantiation that exploits this structure. Third, we show that the problem of verifying decision-tree policies is **PSPACE**-complete even for very simple (namely, state-independent) dynamics. Finally, we demonstrate that our algorithm can verify several reinforcement-learning benchmarks from classical control and the quadrotor from Figure 1, even for unbounded time.

## 1.1 Related Work

A decision tree is a popular machine-learning model that has recently regained interest due to its high interpretability [Du et al., 2020]. Decision-tree policies can be trained directly from a tabular dataset of state-action pairs [Quinlan, 1996], e.g., using the CART algorithm [Breiman et al., 1984] or dtControl [Ashok et al., 2020]. Since these algorithms perform an equivalence transformation, safety guarantees transfer from the dataset to the decision tree. Ashok et al. [2019] show how to extract a safe-by-design decision tree from a safe policy synthesized with Uppaal Stratego [David et al., 2015] for a priced timed game. However, for nonlinear systems it is generally unclear how to obtain safe-by-design policies. In practice one uses best-effort methods such as reinforcement learning. As these approaches cannot guarantee that the resulting policy is safe, there is need for approaches to verify a given policy after learning.

Decision trees can be trained to imitate another model such as a neural network. For instance, the Viper algorithm incorporates decision-tree learning into the imitation procedure [Bastani et al., 2018]. As this imitation provides no correctness guarantees, a separate analysis is required. The authors encode the bounded-time reachability problem for a discrete-time system in logical constraints. This encoding is the only verification approach for discrete-time DTCS we are aware of. Other verification efforts for decision trees have focused on robustness and adversarial examples in a supervised setting [Urban and Miné, 2021], which are orthogonal problems to our setting of verifying a policy.

For purely dynamical systems (without a control policy), reachability analysis has been studied extensively [Doyen et al., 2018, Althoff et al., 2021]. One prominent approach is based on Taylor models [Berz and Makino, 1998], which we also employ in our implementation. Combining a

dynamical system with a control policy results in a hybrid system [Henzinger, 1996]. Applying tools for general hybrid-system verification to DTCS is not feasible because they do not exploit the specific structure (we discuss this further in Appendix D). The reach-avoid problem is undecidable for nonlinear dynamics (even without a control policy), and most approaches (including ours) consider only time-bounded analysis. However, as we discuss later, under certain conditions, our results still generalize to unbounded time via fixpoint techniques [Giacobbe, 2019, Bacci et al., 2021]. For piecewise-constant dynamics, axis-aligned policies resemble systems that can be efficiently dealt with using interval-based arithmetic constraint solving [Fränzle et al., 2007, Fränzle and Herde, 2007].

Closely related to DTCS are neural-network controlled systems (NNCS), where the policy is implemented by a neural network. Recently, many reachability approaches have been proposed [Dutta et al., 2019, Tran et al., 2020, Fan et al., 2020, Akintunde et al., 2022, Ivanov et al., 2021, Schilling et al., 2022, Kochdumper et al., 2023], and implementations compete in a yearly competition [Lopez et al., 2022]. The main difference to DTCS is that the control action comes from a continuous domain and the neural network implements a smooth function. Hence such policies typically yield a similar control action for similar states, which benefits set-based reachability analysis. In contrast, given two similar states, a decision tree can yield vastly different control actions. Thus tools for verifying NNCS are not applicable to our problem. NNCS are Turing-complete [Hyötyniemi, 1996]. We show the same result for DTCS even with the simplest environment dynamics, which yields an undecidable problem in unbounded time and a **PSPACE**-complete problem in bounded time.

To summarize, decision-tree policies are highly desirable due to their interpretability. The most successful methods in obtaining such policies use machine learning, which do not guarantee safety of the resulting policy. While previous work managed to verify discrete-time policies, continuous-time policy verification remains an open problem. Our work bridges this gap with the first algorithm to verify decision-tree policies for continuous-time systems.

**Outline.** We organize the remainder of the paper as follows. First we formalize DTCS and the reach-avoid problem. Then we describe our verification algorithm and discuss the problem complexity. Next we report on experimental results. Finally we conclude and discuss directions for future work.

## 2 Preliminaries

### 2.1 Decision-Tree Controlled Systems

Let $\mathcal{S} \subseteq \mathbb{R}^n$ be an $n$-dimensional state space and $\mathcal{U} \subseteq \mathbb{R}^m$ an $m$-dimensional action (or input) space. A *decision tree* $\mathcal{T}$ over $\mathcal{S}$ and $\mathcal{U}$ is a binary tree such that each inner node is labeled with a predicate $p : \mathcal{S} \to \mathbb{B}$ (with $\mathbb{B} = \{\top, \bot\}$) and each leaf is labeled with an action $u \in \mathcal{U}$. The nodes in the tree are organized in levels, with the root node being at level 1. Let $root(\mathcal{T})$ denote the root node of $\mathcal{T}$, $left(\mathcal{T})$ resp. $right(\mathcal{T})$ denote the left resp. right sub-tree of $\mathcal{T}$, and $\ell(\mathcal{T})$ denote the label at the root of $\mathcal{T}$ (i.e., if $\mathcal{T}$ is a leaf, $\ell(\mathcal{T})$ is an action and otherwise a predicate). We can interpret $\mathcal{T}$ as a function from $\mathcal{S}$ to $\mathcal{U}$ as follows. Given a state $x \in \mathcal{S}$, the image under $\mathcal{T}$, written $act(x, \mathcal{T}) \in \mathcal{U}$, is defined recursively. For a leaf $\mathcal{T}$, $act(x, \mathcal{T})$ is just $\ell(\mathcal{T})$. For a proper tree with root predicate $p$, $act(x, \mathcal{T})$ is $act(x, left(\mathcal{T}))$ if $p(x) = \top$, and $act(x, right(\mathcal{T}))$ otherwise. In this paper, we restrict our analysis to predicates of the form $x_i \leq c$ where $x_i$ is the $i$-th state and $c \in \mathbb{R}$ is a constant. Geometrically, these predicates are axis-aligned half-spaces. This class of predicates is commonly used, e.g., in the tools Uppaal Stratego [David et al., 2015] (with industrial applications in control of smart homes [Larsen et al., 2016] and traffic lights [Eriksen et al., 2017]) and dtControl [Ashok et al., 2020].

We consider environments modeled by a system of ordinary differential equations (ODEs), $\dot{x} = f(x, u)$, where $x(t) \in \mathcal{S}$ is the state vector and $u(t) \in \mathcal{U}$ is the vector of control actions. Given an initial state $x(0) = x_0$ and an action $u_0$, we assume that the solution to the corresponding initial-value problem at time $t \geq 0$, written $\xi(x_0, u_0, t, f)$, exists and is unique (e.g., by Lipschitz continuity).

A *decision-tree controlled system* (*DTCS*) is a triple $(f, \mathcal{T}, \tau)$ where $f$ describes a system of ODEs, $\mathcal{T}$ is a decision tree, used as policy, and $\tau \in \mathbb{R}^+$ is a control period. Figure 2 shows a conceptual sketch. The DTCS periodically queries the policy for a new control action. At time points $k\tau$, $k \in \mathbb{N}$, the current state $x(k\tau)$ is passed to the policy $\mathcal{T}$, which instantaneously yields the new control action $u_k$; this action is then used for the next control period $\tau$ in the system dynamics $f$. Formally, given an initial state $x_0$ at $t = 0$, we recursively define the sequence of actions $u_k = act(x(k\tau), \mathcal{T})$, $k \in \mathbb{N}$, and the evolution of the state $x(t) = \xi(x(k\tau), u_k, t - k\tau, f)$ (i.e., a trajectory) for $t \in (k\tau, (k+1)\tau]$.

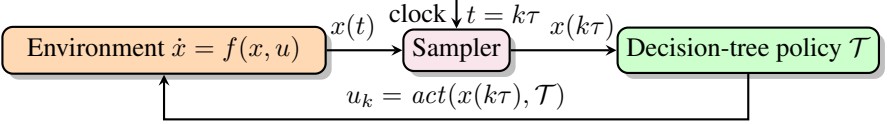

Figure 2: A decision-tree controlled system $(f, \mathcal{T}, \tau)$.

## 2.2 Reachable States and Reach-Avoid Problem

Given a DTCS and a set of initial states $\mathcal{X}_0 \subseteq \mathcal{S}$, we are interested in the *reachable states*, either at time $t$ as $\mathcal{R}_t = \{x(t) \mid x(0) \in \mathcal{X}_0\}$ or the generalization to time intervals $\mathcal{R}_{[T_0, T_1]} = \bigcup_{t \in [T_0, T_1]} \mathcal{R}_t$.

Similarly, a *discrete-time DTCS* is a pair $(f, \mathcal{T})$ where $f$ describes a recurrence $x_{k+1} = x_k + f(x_k, u_k)$ and $\mathcal{T}$ is a decision tree. The reachable states at step $k$ are $\mathcal{R}_k = \{x_k \mid x_0 \in \mathcal{X}_0\}$ and analogously $\mathcal{R}_{[K_0, K_1]} = \bigcup_{k \in [K_0, K_1]} \mathcal{R}_k$. By default, *DTCS* refers to the *continuous-time DTCS* introduced above.

The *reach-avoid problem* for DTCS is, given a DTCS $(f, \mathcal{T}, \tau)$ over $\mathcal{S}$, a set of initial states $\mathcal{X}_0 \subseteq \mathcal{S}$, a bounded number of control cycles $k_{\max}$, and two sets $\mathcal{G}, \mathcal{E} \subseteq \mathcal{S}$, to decide whether all trajectories $x(t)$ reach the goal set $\mathcal{G}$ without reaching the error set $\mathcal{E}$ before time $T_{\max} = k_{\max}\tau$, i.e., $\exists t^* \leq T_{\max}.\ x(0) \in \mathcal{X}_0 \wedge x(t^*) \in \mathcal{G} \wedge \forall t \in [0, t^*].\ x(t) \notin \mathcal{E}$. If we assume that the goal states $\mathcal{G}$ are absorbing, this is equivalent to checking

$$\mathcal{R}_{T_{\max}} \subseteq \mathcal{G} \wedge \mathcal{R}_{[0, T_{\max}]} \cap \mathcal{E} = \emptyset. \tag{1}$$

**Example 2.** *Consider again the quadrotor model from Figure 1. A full model description is given in Appendix B.1. The reach-avoid problem here consists of the shaded area at the top as the goal set $\mathcal{G}$, the set of states outside the red corridor as the error set $\mathcal{E}$, and $k_{max} = 30$ control cycles.*

Determining reachability is already undecidable for uncontrolled nonlinear dynamical systems [Hainry, 2008], and hence also for DTCS with nonlinear dynamics. A DTCS can be seen as a hybrid (mixed discrete-continuous) system, for which reachability is undecidable even with linear dynamics [Doyen et al., 2018]. Due to these complexity barriers, we aim at *enclosing*, or overapproximating, the reachable states up to time horizon $T_{\max}$ by computing a set $\overline{\mathcal{R}} \supseteq \mathcal{R}_{[0, T_{\max}]}$.

## 3 Approach

**Algorithm 1** Reachability algorithm for DTCS

**Input:** DTCS $(f, \mathcal{T}, \tau)$, initial set $\mathcal{X}_0$, time/cycle bounds $T_{\max} = k_{\max} \cdot \tau$
**Output:** set of states $\overline{\mathcal{R}} \supseteq \mathcal{R}_{[0, T_{\max}]}$

1: $\overline{\mathcal{R}} \leftarrow \mathcal{X}_0$
2: $Q \leftarrow \{(\mathcal{X}_0, 0)\}$
3: **while** $\neg$ isempty($Q$) **do**
4:    $(\mathcal{X}, t_0) \leftarrow \text{pop}(Q)$
5:    **if** $t_0 \geq T_{\max}$ **then**
6:       continue
7:    **end if**
8:    $t_1 = \min(t_0 + \tau, T_{\max})$
9:    **for** $(\mathcal{X}_u, u) \in post_{\mathcal{T}}(\mathcal{X}, \mathcal{T})$ **do**
10:      $(\mathcal{Y}, \mathcal{Z}_{t_1}) \leftarrow$
        $post_f(\mathcal{X}_u, u, f, [t_0, t_1])$
11:      $\overline{\mathcal{R}} \leftarrow \overline{\mathcal{R}} \cup \mathcal{Y}$
12:      push($Q, (\mathcal{Z}_{t_1}, t_1)$)
13:    **end for**
14: **end while**
15: **return** $\overline{\mathcal{R}}$

In this section, we present our approach to reachability analysis for DTCS. We first describe a general high-level algorithm, which resembles standard reachability schemes for hybrid systems. It is, however, tailored to policies over a finite action space $\mathcal{U}$. As such, it is applicable to policies beyond decision trees, but for instance not to typical neural-network policies. We then outline conditions under which the algorithm is sound and relatively complete (i.e., does not introduce additional approximation errors). Finally, we instantiate the algorithm specifically for decision-tree policies, which is a novel contribution of this paper.

Algorithm 1 shows our high-level reachability method, which can be used to solve the reach-avoid problem (Section 2.2). The algorithm is parametric in two procedures, $post_{\mathcal{T}}$ and $post_f$, which together compute an enclosure of the reachable states for one control cycle. The queue $Q$ contains pairs $(\mathcal{X}, t)$, where $\mathcal{X}$ is a set of states that needs to be explored, and $t$ is a time point. Each iteration of the while loop (line 3) analyzes one control cycle from $\mathcal{X}$ and $t$ (unless the time horizon $T_{\max}$ is reached, line 6). The result is added to the set $\overline{\mathcal{R}}$ of reachable states in line 11. Next we describe the requirements for the *post* procedures.

The procedure $post_{\mathcal{T}}$, which is a new contribution of this paper, takes a set of states $\mathcal{X}$ and a decision tree $\mathcal{T}$ and returns a finite set of pairs $(\mathcal{Y}_i^u, u)$, where $\mathcal{Y}_i^u$ is a set of states and $u \in \mathcal{U}$ is an action. Since multiple leaves of a decision tree can be associated with the same action $u$, we allow multiple sets $\mathcal{Y}_i^u$ to be associated with $u$ as well. For each action $u$, the union of the sets $\mathcal{Y}_i^u$ should enclose the set $act_u(\mathcal{X}, \mathcal{T}) = \{x \in \mathcal{X} \mid act(x, \mathcal{T}) = u\}$, so we have an index set $I_u$ (possibly empty) for each action $u$ such that

$$act_u(\mathcal{X}, \mathcal{T}) \subseteq \bigcup_{i \in I_u} \mathcal{Y}_i^u. \tag{2}$$

Procedure $post_{\mathcal{T}}$ wraps the sets $\mathcal{Y}_i^u$ together with action $u$.

$$post_{\mathcal{T}}(\mathcal{X}, \mathcal{T}) = \bigcup_{u \in \mathcal{U}} \bigcup_{i \in I_u} \{(\mathcal{Y}_i^u, u)\} \tag{3}$$

For $post_{\mathcal{T}}$ to be sound, we connect equations (2) and (3) and untangle the pairing with $u$ as follows:

$$\forall u \in \mathcal{U}. \ \bigcup \{x \in \mathcal{Y}_i^u \mid (\mathcal{Y}_i^u, u) \in post_{\mathcal{T}}(\mathcal{X}, \mathcal{T})\} \supseteq act_u(\mathcal{X}, \mathcal{T}) \tag{4}$$

The procedure $post_f$ receives a set of states $\mathcal{X}$, a control action $u$, the environment $f$, and a time interval $[t_0, t_1]$. (Note that we actually only need the duration $t_1 - t_0$, which is $\tau$ most of the time, because we consider time-invariant systems. We only pass the time interval for notational purposes to refer to $t_1$.) The goal is to perform a classical time-bounded reachability computation and return two sets $\mathcal{Y}$ and $\mathcal{Z}$ such that $\mathcal{Y}$ encloses the reachable states for the given time interval and $\mathcal{Z}$ encloses the reachable states at the final time point $t_1$. Thus for $post_f$ to be sound, the following must hold:

$$post_f(\mathcal{X}, u, f, [t_0, t_1]) = (\mathcal{Y}, \mathcal{Z}) \implies \mathcal{Y} \supseteq \mathcal{R}_{[t_0, t_1]} \land \mathcal{Z} \supseteq \mathcal{R}_{t_1} \tag{5}$$

Algorithm 1 in combination with equations (4) and (5) computes a sound enclosure. Let us write $post_{\mathcal{T}}^*$ and $post_f^*$ for the procedures such that the inclusions in equations (4) and (5) are satisfied with equality. With such procedures, the algorithm even produces the exact result (proven in Appendix C.1).

**Theorem 1** (Termination, soundness, relative completeness). *Assume Eq. (4) and Eq. (5) are satisfied. (1) Algorithm 1 terminates if all calls to $post_{\mathcal{T}}$ and $post_f$ terminate. (2) Let $\sqsupseteq \in \{\supseteq, =\}$. The result $\overline{\mathcal{R}}$ of Algorithm 1 encloses ($\supseteq$) resp. equals ($=$) the reachable states, $\overline{\mathcal{R}} \sqsupseteq \mathcal{R}_{[0, T_{max}]}$, if in all steps $post_{\mathcal{T}}(\mathcal{X}, \mathcal{T}) \sqsupseteq post_{\mathcal{T}}^*(\mathcal{X}, \mathcal{T})$ and $post_f(\mathcal{X}_u, u, f, [t_0, t_1]) \sqsupseteq post_f^*(\mathcal{X}_u, u, f, [t_0, t_1])$ hold.*

### 3.1 Implementing the Post Procedures

---
**Algorithm 2** Post for the environment

---
**Input:** set $\mathcal{X}$, control action $u$, environment $f$, interval $[t_0, t_1]$
**Output:** pair $(\mathcal{Y}, \mathcal{Z})$ such that $\mathcal{Y} \supseteq \mathcal{R}_{[t_0, t_1]}$ and $\mathcal{Z} \supseteq \mathcal{R}_{t_1}$
1: $P(t) \leftarrow$ TM_reach$(\mathcal{X}, u, f, [t_0, t_1])$
2: $\mathcal{Y} \leftarrow$ evaluate$(P(t), [t_0, t_1])$
3: $\mathcal{Z} \leftarrow$ evaluate$(P(t), t_1)$
4: **return** $(\mathcal{Y}, \mathcal{Z})$

---

For implementing $post_f$, we use an algorithm based on Taylor models [Berz and Makino, 1998] for reachability analysis of nonlinear dynamical systems. A Taylor model approximates a function as a polynomial together with an interval remainder over a domain [Makino and Berz, 2003], which we interpret as sets of states. For example, the one-dimensional Taylor model with polynomial $p(x) = x^2 - x + 1$ and remainder $[-0.5, 0.5]$ over the domain $[-1, 1]$ around expansion point 0 encodes the set $\{p(x) + r \mid x \in [-1, 1], r \in [-0.5, 0.5]\}$. Taylor models subsume common set representations. Thus we assume that the initial set $\mathcal{X}_0$ is given as a Taylor model.

Algorithm 2 summarizes the implementation of $post_f$. Here TM_reach$(\mathcal{X}, u, f, [t_0, t_1])$ (line 1) computes a special Taylor model $P(t)$ that depends on time $t$. To obtain $\mathcal{Y}$ and $\mathcal{Z}$, we evaluate this Taylor model with different values for $t$: For $\mathcal{Y}$ we evaluate with the time interval $[t_0, t_1]$, which results in an enclosure of $\mathcal{R}_{[t_0, t_1]}$. For $\mathcal{Z}$ we evaluate with the time point $t_1$, which results in an enclosure of $\mathcal{R}_{t_1}$. Algorithm 2 can be adapted for discrete-time systems, where the procedure TM_reach is replaced appropriately, line 2 is skipped, and the result is $(\mathcal{Z}, \mathcal{Z})$.

**Proposition 1.** *Algorithm 2 implements procedure $post_f$ satisfying Eq. (5).*

Algorithm 3 instantiates $post_{\mathcal{T}}$. Recall that the goal of this procedure is to compute enclosures of $act_u(\mathcal{X}, \mathcal{T})$, which are the sets of states that result in an action $u$. Given a set $\mathcal{X}$ and a decision tree $\mathcal{T}$, the idea is to propagate $\mathcal{X}$ down the branches that the states $x \in \mathcal{X}$ would take. While the algorithm is

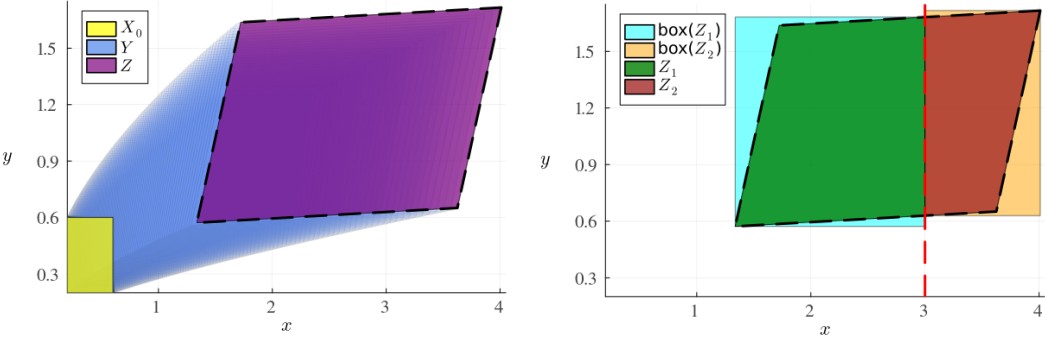

(a) Enclosures of the reachable states $\overline{\mathcal{R}}_{[0,\tau]} = \mathcal{Y}$ (blue) and $\overline{\mathcal{R}}_\tau = \mathcal{Z}$ (purple) starting from $\mathcal{X}_0$ (yellow).

(b) Bisection wrt. $x \leq 3$ (red) into $\mathcal{Z}_1$ (green) and $\mathcal{Z}_2$ (brown), and interval enclosures (cyan, orange).

Figure 3: Example execution of Algorithm 1 for the first control cycle.

agnostic of the predicates in $\mathcal{T}$ (interpreted as sets of states) on a conceptual level, it is generally difficult to implement the algorithm for arbitrary predicates. In our implementation, we exploit the structure of axis-aligned half-space predicates $P$. Let box$(\mathcal{X})$ be the interval enclosure of $\mathcal{X}$, which is easy to obtain for a Taylor model. In line 9, we check if all states in $\mathcal{X}$ take the left branch of $\mathcal{T}$. We have $\mathcal{X} \subseteq P$ if and only if box$(\mathcal{X}) \subseteq P$. In line 11 we check if all states in $\mathcal{X}$ take the right branch of $\mathcal{T}$. We have $\mathcal{X} \cap P = \emptyset$ if and only if box$(\mathcal{X}) \cap P = \emptyset$. If both conditions fail, $\mathcal{X}$ is bisected into $\mathcal{Z}_1 = \{x \mid x \in \mathcal{X} \cap P\}$ and $\mathcal{Z}_2 = \{x \mid x \in \mathcal{X} \cap P^C\}$ in line 14. Here $P^C$ denotes the complement of $P$, which is also a half-space. Since the above sets are hard to compute, we replace $\mathcal{X}$ by box$(\mathcal{X})$ in the implementation. Note that box$(\mathcal{X}) \cap P \supseteq$ box$(\mathcal{X} \cap P)$ in our setting. We discuss this further in Appendix A.1.

**Proposition 2.** *Algorithm 3 implements procedure post$_\mathcal{T}$ satisfying Eq. (4). Furthermore, if all bisections are exact, Eq. (4) is satisfied with equality.*

Further details relevant for using our method in practice are given in Appendix A.

**Example 3.** *We walk through Algorithm 1 for the first control cycle, illustrated in Figure 3. Figure 3(a) shows how Algorithm 2 computes the pair $(\mathcal{Y}, \mathcal{Z})$, consisting of an enclosure $\overline{\mathcal{R}}_{[0,\tau]}$ of the reachable states up to time point $\tau$ and an enclosure $\overline{\mathcal{R}}_\tau$ of the reachable states at the last time point. Figure 3(b) shows how Algorithm 3 bisects the set $\mathcal{Z}$ along the predicate $x \leq 3$ into sets $Z_1$ and $Z_2$. We furthermore illustrate the approximation with interval enclosures. Assuming this is the only predicate of the decision tree, Algorithm 1 would continue the analysis from these two sets in the next iteration.*

---

**Algorithm 3** Post for the decision tree

---

**Input:** set $\mathcal{X}$, decision tree $\mathcal{T}$
**Output:** set of pairs $(\mathcal{X}_u, u)$
1: $S \leftarrow \{\}$
2: $Q \leftarrow \{(\mathcal{X}, root(\mathcal{T}))\}$
3: **while** $\neg$ isempty$(Q)$ **do**
4:     $(\mathcal{Y}, \mathcal{T}_\mathcal{Y}) \leftarrow$ pop$(Q)$
5:     **if** isleaf$(\mathcal{T}_\mathcal{Y})$ **then**
6:        push$(S, (\mathcal{Y}, \ell(\mathcal{T}_\mathcal{Y})))$
7:     **else**
8:        $P \leftarrow \ell(\mathcal{T}_\mathcal{Y})$
9:        **if** $\mathcal{Y} \subseteq P$ **then**
10:          push$(Q, (\mathcal{Y}, left(\mathcal{T}_\mathcal{Y})))$
11:        **else if** $\mathcal{Y} \cap P = \emptyset$ **then**
12:          push$(Q, (\mathcal{Y}, right(\mathcal{T}_\mathcal{Y})))$
13:        **else**
14:          $(\mathcal{Z}_1, \mathcal{Z}_2) \leftarrow$ bisect$(\mathcal{Y}, P)$
15:          push$(Q, (\mathcal{Z}_1, left(\mathcal{T}_\mathcal{Y})))$
16:          push$(Q, (\mathcal{Z}_2, right(\mathcal{T}_\mathcal{Y})))$
17:        **end if**
18:     **end if**
19: **end while**
20: **return** $S$

---

### 3.2 Generalization to Unbounded Time via Fixpoints

Next we discuss how to employ a set-based fixpoint check; for details we refer to the literature [Giacobbe, 2019]. Roughly speaking, if we do not encounter new states after an iteration, we have found a fixpoint. Algorithm 1 iteratively adds states to $\overline{\mathcal{R}}$ to enclose the sequence $\mathcal{R}_{[k\tau,(k+1)\tau]}$, $k < k_{\max}$ (Section 2.2). Let us view the exploration of the elements $(\mathcal{X}, t)$ in the queue $Q$ as a search tree. First, the search in a node can be terminated if the set $\mathcal{X}$ is contained in the union of the sets in the other nodes (i.e., if a fixpoint has been found). Second, if this condition holds in all branches, we have computed the reachable states in unbounded time. However, due to the discrete nature of

the policy, the time points at which to check for fixpoints are relevant. A sufficient condition is to only compare the sets at the beginning of each control cycle, i.e., $\mathcal{R}_{k\tau} \subseteq \bigcup_{j<k} \mathcal{R}_{j\tau}$ for some $k > 0$. However, when dealing with enclosures, this condition would become $\overline{\mathcal{R}}_{k\tau} \subseteq \bigcup_{j<k} \overline{\mathcal{R}}_{j\tau}$, which is not sufficient. The set $\overline{\mathcal{R}}_{k\tau}$ on the left-hand side must additionally contain the reachable states from all sets $\overline{\mathcal{R}}_{j\tau}$ on the right-hand side. Fortunately, our algorithm satisfies this additional condition.

In our implementation, all sets are (unions of) Taylor models, for which inclusion checking is hard. While we can overapproximate the left-hand side, we would have to underapproximate the right-hand side, which is difficult for Taylor models. However, whenever we bisect sets in $post_{\mathcal{T}}$, we obtain box-shaped sets, for which inclusion checks are easy. Thus, in our implementation, we only perform a fixpoint check after a bisection. We shall see in the evaluation that this is often sufficient in practice.

### 3.3 State-Independent Dynamics and Computational Complexity

We call a control system state-independent if $f(x, u) = u$ for all $x$ (for which $u$ must be $n$-dimensional). This is arguably the simplest possible control system. If the system dynamics are state-independent and $\mathcal{X}_0$ is a polyhedron (i.e., the intersection of linear inequalities), $post_f$ can be implemented exactly and yields a polyhedron [Alur et al., 1996]. Given a polyhedron, $post_{\mathcal{T}}$ returns a union of polyhedra. Under these assumptions, Algorithm 1 can be implemented to compute the exact reachable states and thus satisfy Theorem 1 with equality. Finally, Algorithm 1 can be turned into a decision procedure for the reach-avoid problem (namely: also output $\mathcal{R}_{T_{\max}}$ and check Eq. (1)). Nevertheless, state-independent DTCS can encode Turing machines. Hence the reach-avoid problem is undecidable in unbounded time and **PSPACE**-complete in bounded time (proven in Appendix C.2).

**Theorem 2.** *The reach-avoid problem for both continuous-time and discrete-time DTCS over rational numbers with state-independent dynamics and polyhedral initial, goal, and error sets $(\mathcal{X}_0, \mathcal{G}, \mathcal{E})$ is* **PSPACE***-complete, and undecidable when considering the unbounded-time version.*

Note that Algorithm 1 solves not only the decision problem but computes the full reachable states. Thus it needs to collect all the sets resulting from bisections. If the number of iterations $k_{\max}$ is given in binary, the algorithm is doubly exponential, even if we ignore a potential complexity growth in the set representation. We conjecture that this bound is optimal for the full computation. More precisely, in each iteration, a set may be bisected into several (but at most $\ell$, where $\ell$ is the number of leaves of the decision tree) sets. With $k_{\max}$ control cycles, the algorithm may end up with $\ell^{k_{\max}}$ sets in the end.

## 4 Evaluation

We implemented[1] our method in a tool based on JuliaReach [Bogomolov et al., 2019] for the Taylor-model algorithm (TM_reachin Algorithm 2) [Benet et al., 2019] and LazySets [Forets and Schilling, 2021] for the set computations. We demonstrate our approach on the quadrotor system (Figure 1(a)) and three classical control problems from the Gymnasium [Brockman et al., 2016]: cart/pole, acrobot, and mountain/car. In the following, we use our proof-of-concept implementation to verify and visually explain that, within the given time constraints, the quadrotor safely reaches the end of the corridor, the cart manages to stabilize the pole, the acrobot swings to the

Table 1: Sizes of the decision-tree policies, and corresponding verification times averaged over 10 runs.

| System | Policy $\mathcal{T}$ | | | Verification |
|---|---|---|---|---|
| | nodes | depth | actions | time |
| Quadrotor | 177 | 10 | 8 | 84 sec |
| Cart/Pole | 5 | 2 | 2 | 15 sec |
| Acrobot | 7 | 2 | 2 | 101 sec |
| | 9 | 3 | 2 | 113 sec |
| Mountain/Car | 9 | 3 | 3 | 7 sec |

goal height, and the car reaches the top of the mountain. It is crucial to maintain high precision during the analysis to avoid divergence. The systems illustrate different aspects: Our approach applies to time-dependent policies (quadrotor control), proves infinite-time stability (Section 3.2) (cart/pole), handles transformations (Appendix A.2) (acrobot), and deals with discrete-time dynamics (mountain/car), while preserving high precision.

---

[1]The implementation and experiments are available at `https://github.com/VeriXAI/Safety-Verification-of-Decision-Tree-Policies-in-Continuous-Time`.

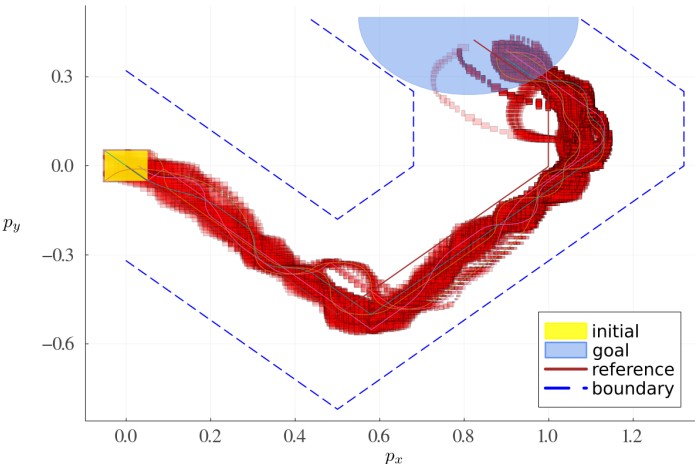

Figure 4: Reachable states (red) and simulations (colored) for the quadrotor system. The initial states are shown in yellow on the left, the goal region is shown in blue at the top, the reference plan is shown in brown, and the dashed blue lines mark the safety corridor.

We obtain decision-tree policies via neural-network distillation (see Table 1 on the depth and number of nodes and actions). For the quadrotor system we collected 500 samples of state-action pairs from a neural-network policy used by Ivanov et al. [2019] and learned a decision tree of depth 10 using behavioral cloning. For the three other systems (cart/pole, acrobot, mountain/car) we first used deep Q-learning to train a two-layer convolutional neural network [Mnih et al., 2013] and then adopted the Viper algorithm [Bastani et al., 2018] to imitation-learn decision trees for these systems. We manually prune the resulting trees of redundant nodes (as Viper produces balanced trees).

All experiments were conducted on a laptop with an i7 1.80 GHz CPU and 32 GB RAM.

### 4.1 Evaluation on a Decision Tree for Quadrotor Control

We evaluate our verification approach on a decision tree controlling a six-dimensional quadrotor, tasked to follow a piecewise-linear plan (see, Example 1). The actions represent possible combinations of pitch, roll, and thrust acceleration. This is a complex continuous-time system (see Appendix B.1 for full details). The policies for such high-dimensional systems are typically approximated by a learned model, e.g., a neural network [Royo et al., 2019]. We train a decision-tree policy of depth 10 imitating the neural network from [Ivanov et al., 2019].

Figure 4 shows the set of reachable states. Ivanov et al. [2019] verified the neural-network policy, for which they had to split the initial set into 16 subsets to tame the approximation error. Computing the reachable states took between 10 and 59 minutes for each subset. Our method can be applied to the full initial set directly and verifies the system in 6.5 minutes.

### 4.2 Evaluation on Decision Trees for Classic Nonlinear Control Problems

Next we study three classical control systems, for which we trained small, interpretable decision trees (see Table 1). Small decision trees are often sufficient for optimality [Vos and Verwer, 2023].

**Cart-Pole System.** We consider the cart-pole system [Barto et al., 1983]. A description is given in Appendix B.2. The goal is for the pole to remain vertically stable within an angle of $\pm 0.06°$. This system is challenging because of quick alternations in the control action. In Figure 5(a), we show the reachable states in the $\theta/\omega$ projection together with the decision boundaries (e.g., the policy moves the cart to the left in the green region). The analysis terminates in 40 seconds and proves that the blue dashed line $\theta = 0.06$ is not exceeded. Furthermore, the fixpoint check allows to generalize the results to infinite time; thus we can conclude that the policy is able to balance the pole forever.

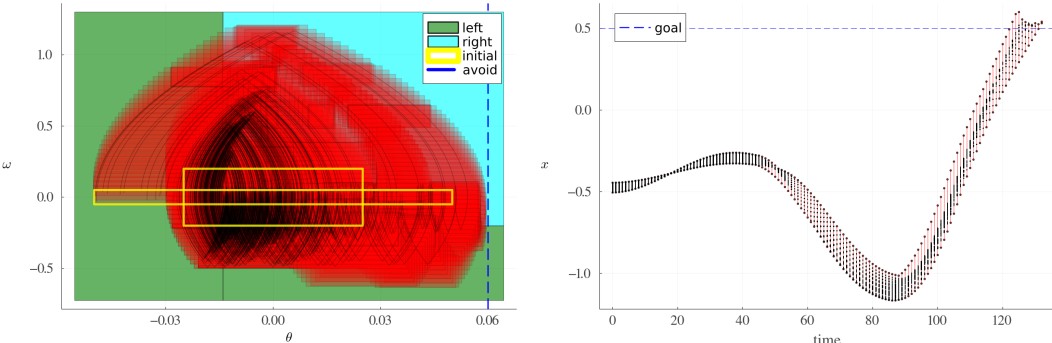

(a) Cart-pole system. The initial states are shown as yellow borders. The angle safety threshold is shown as a dashed blue. The background (green, cyan) shows the decision boundaries.

(b) Mountain-car system. The goal height is shown as a dashed blue.

Figure 5: Reachable states (red) and simulations (black) for the cart-pole and mountain-car systems.

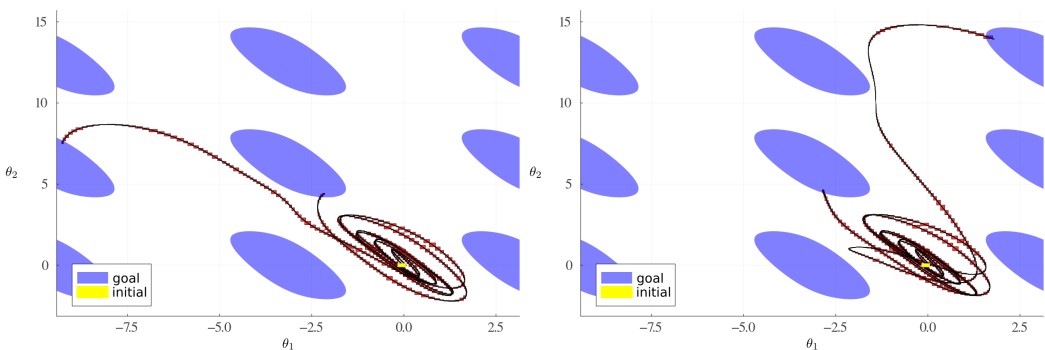

Figure 6: Reachable states (red) and simulations (black) of the acrobot system. The goal regions are shown in blue and the initial states around the origin are shown in yellow (magnified to make them visible). The left (right) plot shows the results for the depth-2 (depth-3) decision-tree policy.

**Acrobot.**   We consider the acrobot system [Sutton, 1995], which consists of two links connected by a joint, one of which can swing freely. A full description is given in Appendix B.3. The goal is that the free end reaches a desired height, expressed as the condition $-\cos(\theta_1) - \cos(\theta_2 + \theta_1) > 1$, where $\theta_1$ is the joint angle and $\theta_2$ is relative to the angle of the first link. This system has complex nonlinear dynamics that require high precision. We obtain two decision-tree policies and compare their performance. The system applies a nonlinear transformation to the state before passing it to the policy. We discuss handling transformations in Appendix A.2. Figure 6 illustrates that all trajectories reach the goal regions. The analysis takes 100 (first policy) resp. 112 seconds (second policy).

**Mountain-Car System.**   To demonstrate applicability to discrete-time systems, we consider the mountain-car system, where a car has to reach from one mountain to another [Moore, 1990]. A description is given in Appendix B.4. Figure 5(b) shows that the car always reaches the top of the mountain within at most 132 steps. The analysis terminates in 16 seconds.

### 4.3   Comparison with State-of-the-Art Reachability Tool for Hybrid Automata

As explained before, a DTCS can be seen as a special case of a hybrid automaton. We encoded the various DTCS from the evaluation as hybrid automata (a description of the encoding is given in Appendix D) and then applied the state-of-the-art reachability tool JuliaReach [Bogomolov et al., 2019] to them. The result was that the tool got stuck as soon as it had to bisect the set of states for the first time, and we had to terminate the tool after several minutes of no progress. For instance, when we evaluated the reachability tool on the cart-pole system, the tool can analyze the first four control periods (out of 25) in 22 seconds, but then gets stuck in the fifth control period when the bisection

starts, and prints errors about numerical instability. This demonstrates that our algorithm is the first *feasible* solution to the reach-avoid problem for DTCS.

Our algorithm exploits the decision-tree structure to only explore the feasible automaton transitions (most of the time this is just one). Furthermore, given the periodic control policy, we only have to check the transitions at specific time points (an observation we already made in earlier work [Forets et al., 2020]). Moreover, since we focus on axis-aligned predicates, interval enclosures are often sufficient. Finally, often only one branch of the decision tree is relevant, which allows us to ignore the complement branch and avoid unnecessary calculations. Even if several leaves of the decision tree are reached, these may still all be annotated with the same control action. In that case we do not have to bisect the set (which avoids the main source of approximation error).

The other tool views the automaton as a black-box model and thus cannot make use of this structure; instead it needs to perform many intersection operations, which are expensive and typically force to use a more complex set representation. These structural insights make our analysis not only more efficient but also more precise in practice because general hybrid-automata tools would typically approximate these operations.

## 5 Conclusion

In this paper, we studied the reach-avoid problem for continuous-time and discrete-time dynamical systems controlled by a decision tree. The problem is undecidable for nonlinear systems, and we showed undecidability even for the simplest dynamics as well as **PSPACE**-completeness in the bounded-time setting. We proposed the first practical algorithm to solve this problem in continuous time. The abstract algorithm is sound and, for simple systems, complete. We implemented the algorithm for nonlinear systems and decision trees with axis-aligned predicates. Our evaluation shows that the algorithm is precise and performant on typical problems. Our approach enriches the verification toolset for machine-learning based systems and opens novel cross-community research challenges. Our approach lends itself to visualization and can serve for further analysis and refinement of decision trees, which themselves are interpretable surrogates for black-box policies.

Coming to the limitations of our approach, we have only considered time-invariant systems. While the algorithms are general, the fixpoint check we used does not apply to time-varying systems. In our experiments, with the exception of the quadrotor, we focused on small decision trees. We note that the size of the decision tree is not necessarily a good measure for complexity. Multiple leaves may share the same action, and some leaves may not be used at all during the execution. The main impact on the verification method, in our experience, is the number of times the decision boundaries are partially crossed by the reachable states (after a time step). Furthermore, the control systems we verified in the evaluation, while nontrivial from a verification perspective, have relatively simple control tasks where a decision tree is not required. We plan to investigate how our method performs on systems with more challenging control tasks, e.g., the cart/pole system starting with the pole hanging downward (which we were not able to verify in a preliminary attempt).

In future work we will also study how the analysis can be improved both in terms of precision and scalability. For precision, we plan to employ set representations that are closed under bisection, e.g., constrained polynomial zonotopes [Kochdumper and Althoff, 2023], which generalize Taylor models. For scalability, we aim to improve the fixpoint check using simulation-based heuristics. Another natural direction is to apply the approach to learning a safe decision-tree policy, for which methods to compute underapproximations in order to refute unsafe models would be useful. Finally, our recent algorithm to synthesize a safety shield for policies of systems with complex hybrid dynamics [Brorholt et al., 2023] only detects discrete-time safety violations; hence we aim to find synergies with the method presented in this paper.

**Acknowledgements.** We are grateful for the anonymous reviewers' helpful comments to improve the paper and suggesting the swing-up cart/pole experiment as future challenge. Anna Lukina thanks Mustafa Mert Çelikok and Alexandru Băbeanu for their timely feedback. This research was partly supported by DIREC - Digital Research Centre Denmark under reference number 9142-0001B and the Villum Investigator Grant S4OS under reference number 37819.

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

# A  Additional Information on our Method

This section contains some technical remarks on the precision of our method as well as on how to deal with state transformations. The latter is relevant for solving one of the benchmark problems in the evaluation (Section 4).

## A.1  Precision of the Algorithm

The precision of the Taylor-model implementation of $post_f$ (Algorithm 2) is good in practice most of the time. The main approximation error in Algorithm 1 comes from the bisections in $post_\mathcal{T}$ (Algorithm 3) and the subsequent approximation, as exemplified in Figure 3(b). This is hard to avoid with nonlinear dynamics, for which bisection of non-convex sets is challenging. With iterative algorithms like ours, the issue with approximation errors is that they accumulate over time, which is known as the wrapping effect [Neumaier, 1993], especially if there are frequent changes of the control actions (i.e., crossings of the decision boundary). To reduce the likelihood of a bisection and accumulation of errors, one can subdivide $\mathcal{X}_0$ into smaller subsets and run an analysis from each of them so that either all or none of the states cross the decision boundary at the same step (for instance, as mentioned in Section 4.1, this idea was used by Ivanov et al. [2019] for the quadrotor model). This can sometimes be achieved more efficiently by only considering the set boundaries [Xue et al., 2017].

## A.2  State Transformations Before Evaluating the Decision Tree

So far we have assumed that the decision tree $\mathcal{T}$ takes the current state as input, as shown in Figure 2. However, some policies instead receive a transformation of the state. This is particularly useful for decision-tree policies with linear predicates (like ours) because a nonlinear transformation effectively augments them with nonlinear predicates. For instance, the acrobot model (see Appendix B.3 for more information) has the state vector $(\theta_1, \theta_2, \dot{\theta}_1, \dot{\theta}_2)$, but the input to $\mathcal{T}$ is the vector $(\cos(\theta_1), \sin(\theta_1), \cos(\theta_2), \sin(\theta_2), \dot{\theta}_1, \dot{\theta}_2)$. To account for transformations, we can modify Algorithm 3 as follows. First we apply the transformation to the input set $\mathcal{X}$, which is easy if $\mathcal{X}$ is given as a Taylor model. Then we apply Algorithm 3, and finally we have to "undo" the transformation for the output. We distinguish three cases. In the first case, if the output is a single pair $(\mathcal{X}_u, u)$ (i.e., there was no bisection), we replace it with $(\mathcal{X}, u)$. In all other cases, there was a bisection. If the transformation is invertible, as in the above case, we apply the inverse transformation (here: $(\arccos(x_1), \arcsin(x_2), x_5, x_6)$, where $x$ is a state of the output set $\mathcal{X}_u$) to each set $\mathcal{X}_u$ in the output pairs $(\mathcal{X}_u, u)$. Otherwise, if the transformation is not invertible, we conservatively replace each pair $(\mathcal{X}_u, u)$ with the pair $(\mathcal{X}, u)$.

# B  Additional Information on the Benchmark Systems

Below we provide further information on the systems from the evaluation (Section 4).

## B.1  Quadrotor System

A quadrotor is tasked to follow a piecewise-linear reference trajectory via bang-bang control (see Figure 1(a)). The six-dimensional state space consists of two 3D vectors for position and velocity. The action space consists of a 3D vector representing possible combinations of pitch ($\theta$), roll ($\phi$), and thrust acceleration ($\alpha$). Using the gravity constant $g = 9.81$, the continuous dynamics are:

$$\dot{p}_x = v_x \qquad \dot{p}_y = v_y \qquad \dot{p}_z = v_z \qquad \dot{v}_x = g\tan(\theta) \qquad \dot{v}_y = -g\tan(\phi) \qquad \dot{v}_z = \alpha - g$$

The policy's task is to minimize the difference between the reference and the actual trajectory. The set of initial states is $\mathcal{X}_0 = [-0.05, 0.05]^2 \times \{0\}^4$. The control period is 0.2 time units and the number of control cycles is 30. The decision-tree policy $\mathcal{T}$ has depth 10 and is too large to depict here.

## B.2  Cart-Pole System

A pole is vertically attached on top of a cart that can be moved left or right along a frictionless track. The goal is to move the cart such that the pole is kept in an upright pose (i.e., the pole angle is small).

The state consists of the cart position $p$, cart velocity $v$, pole angle $\theta$, and angular velocity $\omega$. The action space is $\mathcal{U} = \{-1, 1\}$ ("left", "right") with control variable $u$. The continuous dynamics are:

$$\dot{p} = v \qquad \dot{v} = \psi - \frac{1}{22}\phi\cos(\theta) \qquad \phi = \frac{9.8\sin(\theta) - \cos(\theta)\psi}{2/3 + 5/11\cos(\theta)^2}$$

$$\dot{\theta} = \omega \qquad \dot{\omega} = \phi \qquad \psi = \frac{10u + 0.05\omega^2\sin(\theta)}{1.1}$$

The set of initial states is the union of two hyperrectangles $\mathcal{X}_0 = [-0.05, 0.05]^4$ and $\{0\}^2 \times [-0.025, 0.025] \times [-0.2, 0.2]$. (We added the second hyperrectangle to make the problem more challenging.) The control period is $0.02$ time units. The decision-tree policy $\mathcal{T}$ is shown in Figure 7(a) and works as follows. If the pole is leaning to the left ($\theta \leq -0.014$) or moving in the left direction ($\omega \leq -0.201$), the cart is moved to the left (action $-1$ in the leaves), and otherwise to the right.

The goal is to keep the pole angle $\theta$ within a small range around 0. Figure 8 shows simulations from the corners of $\mathcal{X}_0$ plus 100 random initial states inside $\mathcal{X}_0$ for 2 time units. From these simulations, we can see that $\theta$ seems to stay in the interval $[-0.06, 0.06]$, but some trajectories come close to the upper bound. In the evaluation (Figure 5(a)) we prove this property for $\mathcal{R}_{[0, 0.5]}$.

The fixpoint check allows to conclude that the policy is able to balance the pole forever. The fixpoint is only found in the $\theta/\omega$ projection, but this is sufficient because the cart's position and velocity are irrelevant for the decisions of the tree.

## B.3   Acrobot System

The acrobot system consists of two links connected by a joint to form a chain. One end is fixed while the joint and the other end can swing freely. There are four state dimensions: angles $\theta_1$ and $\theta_2$, and their velocities $\dot{\theta}_1$ and $\dot{\theta}_2$. The action space is $\mathcal{U} = \{0, 1, 2\}$ ("left", "none", "right") with control variable $u$. The continuous dynamics of the velocities are:

$$\ddot{\theta}_1 = -\frac{d_2\psi + \phi_1}{d_1} \qquad\qquad \ddot{\theta}_2 = \psi$$

where

$$d_1 = m_1 lc_1^2 + m_2(l_1^2 + lc_2^2 + 2l_1 lc_2 \cos(\theta_2)) + I_1 + I_2$$
$$d_2 = m_2(lc_2^2 + l_1 lc_2 \cos(\theta_2)) + I_2$$
$$\phi_2 = m_2 lc_2 g \cos\left(\theta_1 + \theta_2 - \frac{\pi}{2}\right)$$
$$\phi_1 = -m_2 l_1 lc_2 \dot{\theta}_2^2 \sin(\theta_2) - 2m_2 l_1 lc_2 \dot{\theta}_2 \dot{\theta}_1 \sin(\theta_2)$$
$$\qquad + (m_1 lc_1 + m_2 l_1)g\cos\left(\theta_1 - \frac{\pi}{2}\right) + \phi_2$$
$$\psi = \frac{u + \frac{d_2}{d_1}\phi_1 - m_2 l_1 lc_2 \dot{\theta}_1^2 \sin(\theta_2) - \phi_2}{m_2 lc_2^2 + I_2 - \frac{d_2^2}{d_1}}$$

The goal is that the free end reaches a desired height, expressed as the condition $-\cos(\theta_1) - \cos(\theta_2 + \theta_1) > 1$, where $\theta_1$ is the joint angle and $\theta_2$ is relative to the angle of the first link.

The initial condition $\mathcal{X}_0$ is a union of a point and a hyperrectangle $\mathcal{X}_0 = (\{-0.0005\} \times \{0\} \cup [-0.0705, -0.0695] \times [-0.0005, 0.0005]) \times \{0\}^2$. The control period is $0.2$ time units and the time

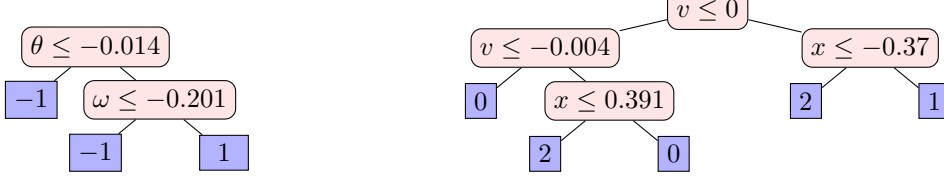

(a) A decision tree for the cart-pole system.          (b) A decision tree for the mountain-car system.

Figure 7: Decision trees for the cart-pole and the mountain-car systems.

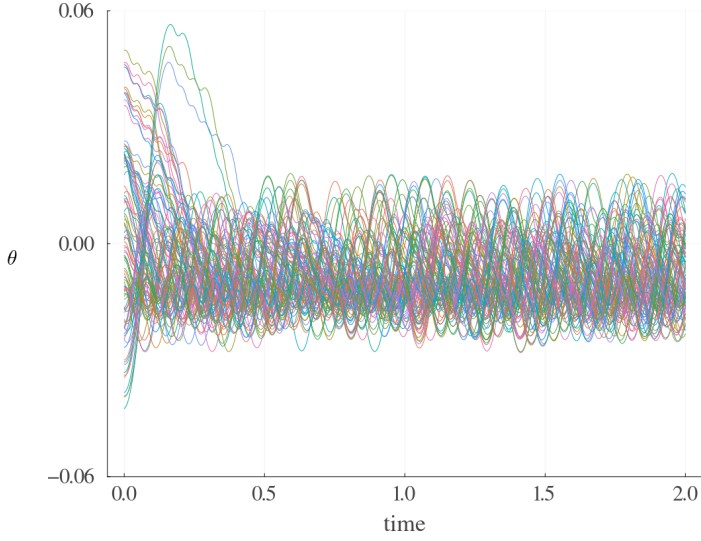

Figure 8: Simulations for the cart-pole system.

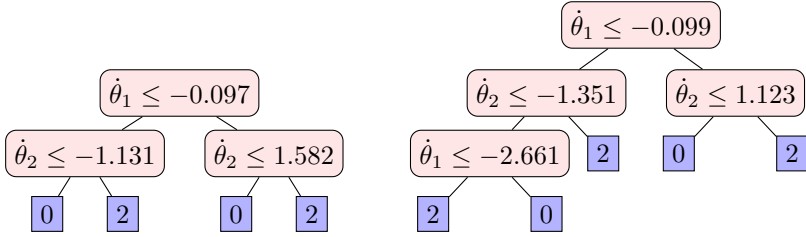

Figure 9: Two alternative decision trees for the acrobot system.

horizon is 100 time units. For this system we obtained two different decision-tree policies of depth 2 resp. 3, shown in Figure 9.

The system applies a transformation to the state before being passed to the decision-tree policy. Concretely, the state vector $(\theta_1, \theta_2, \dot{\theta}_1, \dot{\theta}_2)$ is mapped to the six-dimensional vector $(\cos(\theta_1), \sin(\theta_1), \cos(\theta_2), \sin(\theta_2), \dot{\theta}_1, \dot{\theta}_2)$. (This transformation is technically irrelevant for the two policies we obtained, because they only consider $\dot{\theta}_1$ and $\dot{\theta}_2$, which can be read from the transformed vector; but we do not exploit that.) We explain how to modify our algorithm in order to cope with transformations in Appendix A.2.

### B.4 Mountain-car system

The mountain-car system consists of a sinusoidal valley between two mountains and a car in between. The goal is to bring the car to the top of the higher mountain on the right. The engine is not strong enough to drive up the mountain by itself, so the car must drive up the left mountain and gain momentum to reach the top of the right mountain.

The control actions are $\mathcal{U} = \{0, 1, 2\}$ ("left", "none", "right") with control variable $u$. The system state consists of the car's position $x$ and velocity $v$. With constants $F = 0.001$ and $g = 0.0025$, the discrete-time dynamics are:

$$v_{k+1} = v_k + (u - 1)F - \cos(3x_k)g$$
$$x_{k+1} = x_k + v_{k+1}$$

The set of initial states is $\mathcal{X}_0 = [-0.505, -0.445] \times \{0\}$. The decision-tree policy $\mathcal{T}$ is given in Fig. 7(b). The maximum number of discrete steps is 200, but as shown in Fig. 5(b), all trajectories reach the goal region within at most 132 steps.

# C Proofs of the Theorems

## C.1 Proof of Theorem 1

*Proof.* We show (i) termination and (ii) soundness and relative completeness of Algorithm 1.

(i) The inner loop in line 9 terminates by assumption. The outer loop in line 3 removes one element at time point $t_0$ from $Q$ and adds only elements at time point $t_1 = \min(t_0 + \tau, T_{\max}) > t_0$ to $Q$ (line 8). Elements after time point $T_{\max}$ are not processed (line 6).

(ii) We only show the equality case; the inclusion case follows because, in each step, all sets are overapproximated. The algorithm starts with $\mathcal{X}_0 = \mathcal{R}_0$. In the first iteration, by assumption, $post_{\mathcal{T}}$ only returns states $x_0 \in \mathcal{X}_0$ together with the corresponding action $act(x_0, \mathcal{T})$. Similarly, $post_f$ only returns states reachable within $[0, \tau]$. By the end of the first while-loop iteration, we thus obtain $\overline{\mathcal{R}} = \mathcal{R}_{[0,\tau]}$, and the queue $Q$ contains pairs $(\mathcal{X}_u, \tau)$ such that the union of the sets $\mathcal{X}_u$ is the set $\mathcal{R}_\tau$.

Further loop iterations work similarly. While the iteration order of $Q$ (line 4) is not prescribed, all policies lead to the same result. Let us choose the implementation of $Q$ as a LIFO (last-in, first-out) queue. This corresponds to a breadth-first search where on level $k + 1$ of the search tree we have elements $(\mathcal{X}_i, k\tau)$; the union of these sets $\mathcal{X}_i$ is again $\mathcal{R}_{k\tau}$. The claim follows by induction. $\square$

## C.2 Proof of Theorem 2

We first prove PSPACE-completeness in bounded time. For the membership, we sketch a nondeterministic algorithm with polynomial space requirements for the complement of the reach-avoid problem, which is sufficient by Savitch's theorem [Savitch, 1970] and because **PSPACE** is closed under complementation. The algorithm guesses an initial state and simulates the DTCS. In continuous time, the trajectory in one control cycle is a line segment. In each step, the algorithm checks whether (i) the current line segment intersects with the error states $\mathcal{E}$ and does not intersect with the goal set $\mathcal{G}$, or (ii) the final state in the last iteration $k_{\max}$ is outside the goal set $\mathcal{G}$; these checks can again be performed efficiently. The number of iterations may be exponential, but we can maintain a counter with logarithmically many bits.

For the hardness, we show a polynomial-time reduction from the word problem for deterministic linear bounded automata (LBA). Here we consider the LBA model with accepting state, total transition function, and the input tape containing the input word plus two bound markers for simplicity.

For better intuition, we provide an example after the proof, which we recommend reading in parallel.

Given is an input word $w \in \{0, 1\}^n$ and an LBA $\mathcal{A} = (Q, A, \delta, q_0, q_{\mathrm{acc}}, \#_l, \#_r)$ with states $Q = \{q_0, q_1, \dots\}$, tape alphabet $A = \{a_0, a_1, \dots\} \supseteq \{0, 1, \#_l, \#_r\}$, transition function $\delta \subseteq Q \times A \to Q \times A \times \{L, R\}$ (where $L$ and $R$ denote moving to the left resp. right), initial state $q_0$, accepting state $q_{\mathrm{acc}}$, and bound markers $\#_l, \#_r$.

We define two bijections $N_q : Q \to \{0, \dots, |Q| - 1\}$ and $N_a : A \to \{0, \dots, |A| - 1\}$ to associate a natural number with each state and tape symbol such that $q_{\mathrm{acc}}$ is associated with $|Q| - 1$.

We construct a DTCS with $n + 4$ state variables $x_0, \dots, x_{n+1}, x_q, x_p$, $n + 4$ control actions $u_0, \dots, u_{n+1}, u_q, u_p$, control domain $\mathcal{U} = \mathbb{Z}^{n+4}$ (and control period $\tau = 1$ for continuous systems). The environment dynamics $f$ are state-independent, i.e., $f(x, u) = u$.

The decision tree is more complex, and we first establish some intuition. The state variables encode the LBA configuration, where the first $n + 2$ dimensions $(x_0, \dots, x_{n+1})$ encode the tape contents (i.e., the value of $x_i$ encodes the tape symbol at position $i$), $x_q$ encodes the current LBA state, and $x_p$ encodes the current position on the tape. The execution of one control cycle corresponds to taking one transition. For that we construct the tree from four gadgets: a gadget to select the current LBA state, a gadget to select the current position on the tape, a gadget to select the current tape symbol, and a gadget to apply the transition function to the current configuration.

The first gadget (LBA state selection) is a degenerated tree only expanding to the right with $|Q| - 1$ decision nodes. On the $k$-th level the predicate is $x_q \le k - 1$. The left leaf node at level $k + 1$ is the sub-tree $\mathcal{T}_{k-1}$ corresponding to the second gadget. On the last layer, the right leaf node is $\mathcal{T}_{|Q|-1}$.

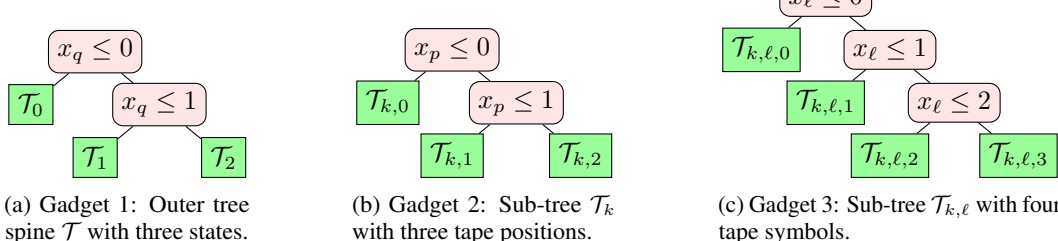

(a) Gadget 1: Outer tree spine $\mathcal{T}$ with three states.

(b) Gadget 2: Sub-tree $\mathcal{T}_k$ with three tape positions.

(c) Gadget 3: Sub-tree $\mathcal{T}_{k,\ell}$ with four tape symbols.

Figure 10: Decision-tree structure from Example 4.

The second gadget (position selection) is an analogous tree $\mathcal{T}_k$ over $x_p$ with $n + 1$ decision nodes. On the $\ell$-th level the predicate is $x_p \leq \ell - 1$. The left leaf node at level $\ell + 1$ is the sub-tree $\mathcal{T}_{k,\ell-1}$ corresponding to the third gadget. On the last layer, the right leaf node is $\mathcal{T}_{k,n+1}$.

The third gadget (symbol selection) is an analogous tree $\mathcal{T}_{k,\ell}$ over $x_\ell$ with $|A| - 1$ decision nodes. On the $m$-th level the predicate is $x_\ell \leq m - 1$. The left leaf node at level $m + 1$ is the sub-tree $\mathcal{T}_{k,\ell,m-1}$ corresponding to the fourth gadget. On the last layer, the right leaf node is $\mathcal{T}_{k,\ell,|A|-1}$.

The fourth gadget (application of the transition function) is a single leave node $\mathcal{T}_{k,\ell,m}$. The indices encode that the LBA is currently in state $N_q^{-1}(k)$ at position $\ell$ and reads tape symbol $N_a^{-1}(m)$. Let us assume that the transition function $\delta$ prescribes to switch to state $q'$, write tape symbol $a'$, and move to the right. Correspondingly, we want to choose the control actions such that in the next step the state of the system is $x_q = N_q(q')$, $x_p = \ell + 1$, $x_\ell = N_a(a')$, and all other state variables remain unchanged. This is achieved by choosing

$$u_q = N_q(q') - k, \quad u_p = 1, \quad u_\ell = N_a(a') - m, \quad u_i = 0 \text{ (for all } i \neq \ell).$$

(For moving to the left, choose $u_p = -1$ instead.) Observe that the state variables keep track of the LBA configuration and that we have corresponding leaves in $\mathcal{T}$ for each transition in $\delta$. For a fixed transition $(q, a, q', a', d)$ we actually have multiple leaves; they only differ in the choice of the control action $u_\ell$ (corresponding to the current cell index on the tape).

Finally, let $w_i$ be the $i$-th symbol of $w$ (starting with $i = 1$). The initial state is $x_q = N_q(q_0), x_p = 1$ (first symbol of $w$), $x_0 = N_a(\#_l)$, $x_{n+1} = N_a(\#_r)$, and $x_i = N_a(w_i)$ for all other $i$. The set of goal states $\mathcal{G}$ consists of all states where $x_q = N_q(q_{\text{acc}}) = |Q| - 1$. The set of error states $\mathcal{E} = \emptyset$ is not needed. The step bound is $k_{\max} = |Q||A|^{n+2}(n + 2)$ (encoded in binary).

The above construction is polynomial in the size of the LBA and the word (the decision tree has $O(|Q| + |A| + n)$ nodes), and the following equivalence shows that it is a reduction.

$$w \text{ is accepted by } \mathcal{A}$$
$$\iff \text{the configuration } \langle q_0, \#_l w \#_r \rangle \text{ leads to a configuration with state } q_{\text{acc}}$$
$$\overset{(*)}{\iff} \text{the DTCS reaches a state with } x_q = N_q(q_{\text{acc}}) \text{ within } k_{\max} \text{ steps}$$
$$\iff \text{the DTCS satisfies the reach-avoid specification}$$

The critical step is $(*)$, for which we rely on the intuition established above. Note that in the continuous-time case we have to make sure that a goal state is not reached accidentally between two control cycles. This is ensured by letting $N_q(q_{\text{acc}}) = |Q| - 1$.

This concludes the proof of **PSPACE**-completeness. Before we continue with the undecidability in unbounded time, we give an example to better illustrate the previous reduction.

**Example 4.** *Consider the LBA with states $q_0, q_s, q_{acc}$, tape symbols $0, 1, \#_l, \#_r$, and the only explicit transition $\delta(q_0, 1) = (q_{acc}, 0, R)$ (all other transitions lead to the sink state $q_s$ with appropriate direction $L/R$). The LBA accepts words starting with a 1 (and changes it to 0). We use the mappings*

$$N_q(q) = \begin{cases} 0 & q = q_0 \\ 1 & q = q_s \\ 2 & q = q_{acc} \end{cases} \qquad N_a(a) = \begin{cases} 0 & a = 0 \\ 1 & a = 1 \\ 2 & a = \#_l \\ 3 & a = \#_r. \end{cases}$$

*Now consider the input word $w = 1$, which means we have the initial configuration $\langle q_0, \#_l 1 \#_r \rangle$. We construct a state-independent DTCS with state variables $x = (x_0, x_1, x_2, x_q, x_p)$ and control actions $u = (u_0, u_1, u_2, u_q, u_p)$. The initial state is $(2, 1, 3, \underbrace{0}_{q_0}, \underbrace{1}_{position})^T$.*

$$\underbrace{2, 1, 3}_{\#_l 1 \#_r}, \underbrace{0}_{q_0}, \underbrace{1}_{position}$$

*The decision tree $\mathcal{T}$ is quite large (36 leaves), and we only depict its structure in Fig. 10. The first cartoon shows the outer structure. Each leaf node follows the structure in the second cartoon, and each leaf node shown there follows the structure in the third cartoon. Finally, we give one example leaf $\mathcal{T}_{0,1,1}$ of the whole tree, which stands for state $q_0$, tape position $1$, and reading symbol $1$. Observe that this is the leaf the decision tree chooses in the initial state. The control-action vector in that leaf is $(\underbrace{0}_{cell\,0}, \underbrace{0-1}_{cell\,1}, \underbrace{0}_{cell\,2}, \underbrace{2-0}_{state}, \underbrace{1}_{position})^T$. Thus, in the next time step, the new state will be $(\underbrace{2, 0, 3}_{\#_l 0 \#_r}, \underbrace{2}_{q_{acc}}, \underbrace{2}_{position})^T$, which is a goal state. This concludes the example.*

Now we prove undecidability in unbounded time. For that, we reduce from the termination problem for deterministic two-counter machines (2CMs). We consider the model from [Shepherdson and Sturgis, 1963, Theorem 4.1] with two counters $c_1, c_2 \in \mathbb{N}$ and instruction set $\texttt{INC}(c_z)$ (increment counter $c_z$), $\texttt{DEC}(c_z)$ (decrement counter $c_z$ with zero lower bound), $\texttt{JZ}(c_z, \ell)$ for $z \in \{1, 2\}$ (jump to location $\ell$ if counter $c_z$ is zero), and $\texttt{STOP}$ (terminate). Each instruction $I$ comes with a position $k$ in the program, which we write "$k : I$". The initial value of the counters is $c_1 = c_2 = 0$ and the program starts at instruction $1$.

Given is a 2CM with $n$ instructions. We construct a DTCS with 3 state variables $x_0, x_1, x_{pc}$, 3 control actions $u_1, u_2, u_{pc}$, control domain $\mathcal{U} = \mathbb{Z}^3$ (and control period $\tau = 1$ for continuous systems). The environment dynamics $f$ are state-independent, i.e., $f(x, u) = u$.

We first establish some intuition. The state variables $x_1$ and $x_2$ correspond to the counters $c_1$ and $c_2$, and the state variable $x_{pc}$ corresponds to the program counter. After each control cycle, these variables will have the corresponding values in the simulated 2CM. For the decision-tree policy, we construct a gadget for each instruction in the 2CM, and one more gadget to select the current instruction based on the value of $x_{pc}$.

We start with the latter gadget (instruction selection), which works precisely as most gadgets in the previous proof. It is a degenerated tree only expanding to the right with $n - 1$ decision nodes. On the $k$-th level, the predicate is $x_{pc} \leq k$. The left leaf node at level $k + 1$ is the sub-tree $\mathcal{T}_k$ corresponding to one of the other gadgets, depending on the $k$-th instruction. The last layer's right leaf node is $\mathcal{T}_n$.

The other gadgets are trees substituted for $\mathcal{T}_k$ depending on the $k$-th instruction. We only describe the instructions for the counter $c_1$; for $c_2$ the construction is completely analogous by swapping 1 and 2 in the indices. For instruction "$k : \texttt{INC}(c_1)$" we set $\mathcal{T}_k$ to the leaf node with $u_1 = 1, u_2 = 0, u_{pc} = 1$. For instruction "$k : \texttt{DEC}(c_1)$" we set $\mathcal{T}_k$ to the tree with decision node $x_1 \leq 0$ and two leaves; the left leaf node is $u_1 = 0, u_2 = 0, u_{pc} = 1$, and the right leaf node is $u_1 = -1, u_2 = 0, u_{pc} = 1$. For instruction "$k : \texttt{JZ}(c_1, \ell)$" we set $\mathcal{T}_k$ to the tree with decision node $x_1 \leq 0$ and two leaves; the left leaf node is $u_1 = 0, u_2 = 0, u_{pc} = \ell - k$, and the right leaf node is $u_1 = 0, u_2 = 0, u_{pc} = 1$. For instruction "$k : \texttt{STOP}$" we set $\mathcal{T}_k$ to the leaf node with $u_1 = 0, u_2 = 0, u_{pc} = -k$.

Finally we define the (singleton) set of initial states $\mathcal{X}_0 = \{(0, 0, 1)\}$, the goal region $\mathcal{G} = \{(x, y, 0) \mid x, y \in \mathbb{R}\}$, and the set of error states $\mathcal{E} = \emptyset$ (not used).

It is easy to see that each of the gadgets implements the 2CM instructions. Executing the action corresponding to the $\texttt{STOP}$ instruction is the only way that $x_{pc}$ can reach the value 0. Thus the above construction is a reduction.

the 2CM terminates
$\Longleftrightarrow$ the $\texttt{STOP}$ instruction is reached from the configuration $c_1 = c_2 = 0, pc = 1$
$\Longleftrightarrow$ the DTCS reaches a state with $x_{pc} = 0$
$\Longleftrightarrow$ the DTCS satisfies the reach-avoid specification $\qquad\qquad\qquad\qquad\quad \square$

## D   Modeling Decision-Tree Controlled Systems as Hybrid Automata

Here we expand on the discussion from Section 4.3 and explain the encoding.

Systems with mixed continuous and discrete dynamics can be modeled as a hybrid automaton [Henzinger, 1996]. As such, DTCS can also be encoded as a hybrid automaton. The reach-avoid problem has been studied extensively for various subclasses of hybrid automata [Doyen et al., 2018, Althoff et al., 2021]. One could thus argue that solutions to the reach-avoid problem for DTCS exist. However, when modeling a DTCS as a hybrid automaton, the problem structure gets lost, and thus tools analyzing this hybrid automaton will not scale. Hence we argue that our algorithm is the first *feasible* solution to the reach-avoid problem for DTCS.

Below we present one possible encoding of a DTCS as a hybrid automaton, for which we assume that the reader is familiar with the terminology of hybrid automata. We encoded the systems from our evaluation (Section 4) accordingly and applied the state-of-the-art reachability tool JuliaReach [Bogomolov et al., 2019] (which also implements an algorithm based on Taylor models [Benet et al., 2019]) to them. The result was that the tool got stuck as soon as it had to bisect the set of states for the first time, and we had to terminate the tool after several minutes of no progress.

We encode a DTCS $(f, \mathcal{T}, \tau)$ as a hybrid automaton as follows. First we introduce a fresh variable $t$ for time. We use one location $\ell_u$ for each control action $u \in \mathcal{U}$ that occurs in $\mathcal{T}$. The continuous dynamics in $\ell_u$ are given by $f(x, v) \wedge \dot{t} = 1$, where we substitute the value $u$ for the second argument $v$. Each location $\ell_u$ has an invariant $t \leq \tau$, restricting time to one control period. For each location there is a transition to every location $\ell_{u'}$ with a guard condition consisting of $t = \tau$ and a big disjunction of conjunctions of the predicates along the paths in $\mathcal{T}$ to leaves with action $u'$. For instance, for $\mathcal{T}$ given in Figure 7(a), the guards on the transitions leading to $\ell_{-1}$ are $t = \tau \wedge (\theta \leq -0.014 \vee (\theta > -0.014 \wedge \omega \leq -0.201))$. Finally, each transition has a reset $t := 0$.

Such an automaton is challenging for black-box analysis tools because they always have to explore all transitions. For instance, when we evaluated the reachability tool on the cart-pole system, the tool can analyze the first four control periods (out of 25) in 22 seconds, but then gets stuck in the fifth control period when the bisection starts, and prints errors about numerical instability. By knowing the problem structure, our approach only explores the feasible paths of the decision tree (often there is just one), for which it only has to evaluate $d$ predicates, where $d$ is the depth of the tree. Furthermore, given the periodic control policy, we only have to perform these checks at specific time points. For general hybrid automata, this has to be found out via intersections with invariants and guards. These structural insights make our analysis not only more efficient but also more precise in practice because general hybrid-automata tools would typically approximate these operations.

