# OpenReview forum: "Safety Verification of Decision-Tree Policies in Continuous Time"
_NeurIPS.cc/2023/Conference — NeurIPS 2023 spotlight_

### Official Review · Reviewer_77nS · 2023-07-06

**Soundness:** 4 excellent
**Presentation:** 3 good
**Contribution:** 3 good
**Rating:** 7
**Confidence:** 5

**Summary:**

This paper introduces a novel verification algorithm that enables the verification of decision tree policies for continuous-time systems (also applicable to discrete-time systems). This approach ensures a sound and compact representation of reachable sets using Taylor models, which can be efficiently propagated through non-linear dynamics systems. The algorithm leverages the decision tree structure to propagate a set-based approximation of abstract reachable states through decision nodes, allowing for splits at decision boundaries. The effectiveness of this approach is demonstrated through the verification of safety for several decision trees that imitate neural-network policies in classic low-dimensional nonlinear control systems (e.g. cartpole). This method can provide robust safety and reachability guarantees for all system behaviors in these benchmarks.

**Strengths:**

+ The proposed algorithm marks a significant advancement as the first formal verification method for verifying reach-avoid properties in a decision-tree controlled system (DTCS) with continuous-time dynamics, including nonlinear systems.
+ It achieves this by efficiently propagating a set of states through the system dynamics and tailoring general hybrid system reachability verification specifically for decision trees. The algorithm effectively utilizes axis-aligned predicates and interval-based set approximations to enhance the precision of abstract states, mitigating the wrapping effect.
+ Moreover, the algorithm showcases its capability to generalize to unbounded time horizons by conducting reachable states fixpoint calculations, providing the potential for inductive safety verification over infinite time horizons.
+ The theoretical analysis conducted in this paper provides insights into the main algorithm.

The paper introduces a tool for ensuring safety and reachability guarantees in decision-tree controlled systems, pushing the boundaries of verification techniques for learning-enabled systems in the context of continuous-time dynamics.

**Weaknesses:**

+ The proposed method consists of two parameterized procedures: one for analyzing dynamical systems and another for decision tree policies. The verification algorithm for dynamical systems is well-established in the related literature. To enhance the clarity of the paper, it would be beneficial to treat the verification procedure as background information in a dedicated section, allowing the paper to primarily focus on its contribution to the analysis of decision tree policy reachability.

+ The evaluation presented in the paper falls short in adequately showcasing the limitations of the verification algorithm in practical settings. To provide a more comprehensive analysis, it would be valuable to explore scenarios where the reachability analysis may potentially diverge. For instance, investigating the scalability of the reachability analyzer to deep decision trees with a significant number of nodes would be helpful. Additionally, conducting an ablation study on decision trees with various shapes may provide insights into the algorithm's behavior.

+ Furthermore, the benchmarks used in the evaluation appear to be simple, and it seems that a linear controller could be trained directly using reinforcement learning for all of these benchmarks and verified using reachability analysis. To strengthen the evaluation, it would be beneficial to include a benchmark scenario where a linear controller fails, thus highlighting the necessity and added value of decision tree controllers that can be additionally verified. For example,  cartpole swinging up could be a suitable benchmark to demonstrate the necessity of decision tree controllers.

**Questions:**

+ Why does the cart-pole benchmark in the paper use a non-standard initial state set? Could the verification algorithm be applied to the cart-pole system using the initial state set provided in the Gym environment?

+ Is the verification algorithm capable of scaling to slightly more complex benchmarks, such as the pendulum and cart-pole swinging up tasks? Can it provide proof for the safety and reachability properties of these systems over an infinite time horizon?

**Limitations:**

The paper lacks a specific discussion on the scalability of the approach in terms of the size and complexity of control systems and decision trees that can be effectively verified. It would be helpful to address this aspect in the final version of the paper.

---

> ### Author Rebuttal · Authors · 2023-08-09
>
> Thank you for the excellent review.
>
> > To enhance the clarity of the paper, it would be beneficial to treat the verification procedure as background information in a dedicated section, allowing the paper to primarily focus on its contribution to the analysis of decision tree policy reachability.
>
> Thank you for the suggestion. We will restructure the paper accordingly.
>
> > The evaluation presented in the paper falls short in adequately showcasing the limitations of the verification algorithm in practical settings. To provide a more comprehensive analysis, it would be valuable to explore scenarios where the reachability analysis may potentially diverge. For instance, investigating the scalability of the reachability analyzer to deep decision trees with a significant number of nodes would be helpful.
>
> Our experiments already include a relatively large decision tree (depth 10, 177 nodes) for the drone quadrotor problem to show that large trees are also admissible, and as we argued in the paper, there exists a generic hybrid-system algorithm to verify DTCS, which is however impractical (Section 4.3). The main message of the paper is that a dedicated verification algorithm significantly outperforms such a generic algorithm.
>
> Nevertheless we agree that scalability is an interesting question, however in reachability analysis evaluating scalability is not straight-forward, since neither does 1) the size of the tree nor 2) the dynamics of the system impact the algorithm in a predictable way. Below we give examples for both points.
>
> 1) The size of the tree is not necessarily dominating because multiple leaves may share the same action, or some leaves may not be used at all during the execution. What matters is the number of times the decision boundaries are partially crossed by the reachable states after each time step. Given that different controllers may implement different policies, we cannot simply compare analyses with controllers of different size, since the verification performance may even improve for larger controllers. For instance, please see the two alternative controllers for the acrobot system (Fig. 9 in the supplementary material), whose behaviors differ significantly (Fig. 6). Note that in general we are interested in smaller trees since this is one of the main motivations for using decision-tree controllers over other (learned) non-interpretable controllers.
>
> 2) It is hard to characterize the 'size' of a dynamical system. For instance, the competition on verifying continuous and hybrid systems (see [1*]) showcases that most verification solvers struggle with the *three-dimensional* Robertson chemical reaction, whereas the *seven-dimensional* Laub-Loomis model has several solvers that easily solve it. Note that this part is orthogonal to our work since our contribution is not in the algorithm for dynamical systems.
>
> We will add this discussion to the paper.
>
> [1*] Geretti et al.: ARCH-COMP22 Category Report: Continuous and Hybrid Systems with Nonlinear Dynamics. 2022.
>
> > it seems that a linear controller could be trained directly using reinforcement learning
>
> Thank you for the suggestion. We understand that this is only a suggestion to strengthen the story. Our acrobot benchmark requires a nonlinear controller. Note that our acrobot is similar to an up-swinging cart pole since the acrobot starts in a downwards position. It is true that some of the benchmarks admit a linear controller, however, obtaining suitable controllers is not the focus of our work.
>
> > Why does the cart-pole benchmark in the paper use a non-standard initial state set? Could the verification algorithm be applied to the cart-pole system using the initial state set provided in the Gym environment?
>
> The set is a union of two sets. The reason we added this set was simply to demonstrate that our approach can naturally deal with non-convex shapes. The first set is similar to the set from the Gym environment. Adding the second set makes the union larger (and hence makes the problem harder).
>
> We re-ran the experiment using the set from the Gym for the first set (i.e., still a harder challenge) and obtained a very similar result as before without any noticeable impact on reachable states and verification time. In the final version, we can use this refined initial set.
>
> > Is the verification algorithm capable of scaling to slightly more complex benchmarks, such as the pendulum and cart-pole swinging up tasks? Can it provide proof for the safety and reachability properties of these systems over an infinite time horizon?
>
> The acrobot benchmark from our evaluation is similar to a pendulum or an up-swinging cart/pole system. Thus we expect that our algorithm may be able to provide a proof, provided the controller is safe and sufficiently stable.

---

> > ### Comment · Reviewer_77nS · 2023-08-15
> > **Thank you for your response**
> >
> > Thanks for the detailed response. I find it difficult to draw a parallel between the acrobot benchmark and the cart pole swing-up problem as they differ significantly in their requirements. The acrobot benchmark only requires reaching a goal position, while the cart pole swing-up problem demands both reaching a vertical position and maintaining balance. The cart pole problem does require a nonlinear controller, but the acrobot task can be tackled with a simple linear one for reachability only. My concern remains about the simplicity of the benchmarks used for evaluation. This raises questions about whether the observed good performance is due to the lack of significant nonlinearity in the learned controllers.
> >
> > The cart pole swing-up problem not only serves as a suitable benchmark to showcase the importance of decision tree controllers but also points to a more difficult verification issue – how can your approach be used to verify controller correctness over an infinite time horizon?
> >
> > I kindly request the authors to acknowledge these limitations in their paper or provide evidence demonstrating that their tool is indeed capable of verifying the full correctness of a cart pole swing-up controller.

---

> > > ### Author Response · Authors · 2023-08-18
> > > **Follow-up on swing-up model**
> > >
> > > Thank you for the clarification. We agree that, while the dynamics of the swing-up cart/pole system and the acrobot system are similar, the control tasks differ.
> > >
> > > We do not see a fundamental reason why nonlinearity of the controller would make the _verification_ task inherently more difficult. Note that decision-tree controllers are nonlinear controllers.
> > >
> > > Regarding your question about the swing-up cart/pole system, we agree that it would be interesting to include this benchmark in addition to the four benchmarks considered in our work. Intuitively it should be possible to verify, given that up-swinging cart/pole becomes a standard cart/pole after the pole is brought into the upward position, and we verified both acrobot and cart/pole.
> > >
> > > We considered experimenting with swing-up cart/pole system during this short rebuttal phase, but did not manage to set it up due to some numerical stability issues that go beyond the contributions of our work (see below for details).
> > >
> > > In the camera-ready version we will add a discussion about the limitations you raised.
> > >
> > > ---
> > >
> > > ### Issues with the implementation of the swing-up cart/pole system
> > >
> > > We had to adapt the differential equations wrt. the model in the paper to faithfully model the swing-up phase. For that, we found the system dynamics online and trained a decision tree that seemingly manages to do the control (based on simulations from selected points, but not verified). However, the new differential equations are more complex than what we used in our cart/pole system, and it turned out that the Taylor-model algorithm struggles with this model. Although the Taylor-model algorithm can maintain precision for most of the up-swing phase, at some point it starts diverging in our implementation.
> > >
> > > The issue with approximation error even arises when starting from a single point - for which no splitting occurs in our algorithm, and thus our algorithm does not add any approximation error. Hence these numerical issues would occur for any algorithm that uses a Taylor-model algorithm for the continuous-time reachability analysis (which is out of scope for our paper).
> > >
> > > We will consider modelling the problem using a simpler set of differential equations that could still reasonably well represent the system, but we did not manage to do this during the rebuttal phase.

---

> > > > ### Comment · Reviewer_77nS · 2023-08-19
> > > > **Thanks for the result**
> > > >
> > > > I appreciate the authors for sharing the new experiment result. I support accepting this paper. I encourage the authors to discuss the limitations and consider exploring pendulum swing-up in the final version of the paper, which is simpler and requires a nonlinear controller.

---

### Official Review · Reviewer_yJkM · 2023-07-06

**Soundness:** 4 excellent
**Presentation:** 4 excellent
**Contribution:** 4 excellent
**Rating:** 7
**Confidence:** 3

**Summary:**

The paper presents an approach for verifying safety (reach-avoid) properties of controlled systems where  the state space of the system is continuous, its dynamics are continuous-time, and the policy/controller is described by a decision tree that chooses actions from a continuous action space. The proposed reachability-based verification algorithm invokes two main sub-procedures $post_f$ and $post_{\mathcal{T}}$. Given an initial set of states and an action, $post_f$ computes the reachable states induced by the dynamics using standard techniques based on Taylor models. On the other hand, $post_{\mathcal{T}}$ uses the structure of a decision tree to compute an overapproximation of all possible state-action pairs induced by the policy given an initial set of states. The design of $post_{\mathcal{T}}$ as well as the overall verification algorithm are the primary contributions of this work. The paper also includes theoretical results establishing the soundness and the relative completeness of the verification algorithm, as well as results establishing the computational hardness of the safety verification problem for systems with decision tree controllers. Moreover, empirical evaluation shows the scalability of the verification algorithm compared to other, more generic techniques.

**Strengths:**

The paper presents an elegant verification algorithm as well as establishes hardness results for an interesting, well-specified family of controlled systems. The overall verification algorithm is clean and generic, and provides precise specifications that need to be satisfied by the sub-procedures $post_f$ and $post_{\mathcal{T}}$ for the overall algorithm to be a sound and complete verifier. The key idea for computing reachability through the decision tree policy---namely, using box abstractions for efficient calculation of inclusion checks and bisection operation---is sensible and intuitive, and suggests clear directions for future research. The paper also presents useful insights on extending the analysis to unbounded time settings. The empirical evaluation supports the claim that designing a verification procedure that exploits the structure of Decision Tree Controlled Systems (DTCS) can lead to efficiency wins. I also greatly appreciate the quality of the writing and the formal rigor of the presented ideas (barring a few concerns described below).

**Weaknesses:**

My primary concerns have to do with the presentation of the technical material and the comparison with related work.

1. The paper repeatedly suggests that exploiting the special structure of DTCS for verification is one of the key insights of the presented work but a clear explanation of this special structure and how it is exploited is only cursorily explained in Section 4.3 (with a more detailed explanation in the appendix). I am not an expert on the topic and for readers like me, it would help if these structural insights were clearly explained in the initial sections of the paper.

2. I find the presented ideas about generalizing to unbounded time via fixpoints very interesting. However, the text in Section 3.2 seemed handwavy, and I would very much appreciate a more precise explanation of the ideas. For instance, I found the comments on line 226 ("However, due to the discrete ...") and on line 230 quite cryptic. I am also confused about the fact that while $post_f$ takes both a set of states and a time interval as input parameters, the fixed point check only considers the input states (i.e., assumes time-invariant dynamics). If the dynamics is time-invariant, why does $post_f$ need $t_0$ and $t_1$ as parameters? Can $t_0$ and $t_1$ be fixed as 0 and $\tau$?

3. The evaluation would be much stronger if there were a head-to-head comparison of the algorithm in this paper with reachability tools for hybrid automata. Section 4.3 briefly touches upon this, but a direct comparison for all four examples would be quite instructive. Another minor comment is that given the amount of detail in Figures 4,5, and 6, it would help to make them higher resolution.

**Questions:**

My questions are related to comments above in the Weaknesses section.

1. What structure of decision trees does the algorithm exploit? Is the structure primarily used in the design of Algorithm 3, or does Algorithm 1 also exploit the structure?

2. Repeating the second comment above, the fixed point check seems to assume time-invariant dynamics. Is this correct? If so, why pass $t_0$ and $t_1$ as parameters to $post_f$ instead of simply passing $\tau$?

**Limitations:**

The paper does not explicitly discuss the limitations and might benefit from a small discussion about the same. For instance, a discussion about the scalability of the approach and loss of precision due to approximations (related to the discussion in the last para of the conclusion) would be helpful.

---

> ### Author Rebuttal · Authors · 2023-08-09
>
> Thank you for the excellent review.
>
> > exploiting the special structure of DTCS for verification is one of the key insights of the presented work but a clear explanation of this special structure and how it is exploited is only cursorily explained.
>
> Thank you for the suggestion. We will improve the explanation. For context, remember that in our setting, Algorithm 3 receives a set of states X, which is then propagated down the branches of the decision tree. Compared to analyzing a DTCS with a generic hybrid-system tool, the most important aspects of the special structure are as follows.
>
> (1) The decision tree partitions the set X. In particular, each predicate either cuts X in two or keeps it intact. Furthermore, if X satisfies a DT predicate, we do not have to explore the complement branch. In a general hybrid system, there is no "else case" and the conditions may overlap; hence the (expensive) intersection computations have to be performed repeatedly and the complement branch has to be analyzed as well.
>
> (2) If the set X only leads to leaves with the same action, we do not need to split it (this happens quite often), which avoids the main source for loss of precision altogether. In a general hybrid system, combined with aspect (1) above, the connection is lost.
>
> (3) Axis-aligned predicates allow to use (simple but efficient) box abstractions.
>
> > Is the structure primarily used in the design of Algorithm 3, or does Algorithm 1 also exploit the structure?
>
> Algorithm 1 does not exploit the structure of decision trees (it is actually generic in the type of controller). But it exploits the structure of a periodic control system: Algorithm 3 only has to be called at fixed points in time, and Algorithm 2 has to be run for a fixed amount of time.
>
> > I find the presented ideas about generalizing to unbounded time via fixpoints very interesting. However, the text in Section 3.2 seemed handwavy
>
> Thank you for the feedback. This section is indeed a very brief overview. The discussion is considered folklore in the reachability community, and the reason we included it is to be self-contained with respect to the experiments. In short, once the system finds itself in a state that was already analyzed previously, we can conclude that the system has reached a fixed point. This observation naturally generalizes to sets of states. We will make the description more formal.
>
> > The evaluation would be much stronger if there were a head-to-head comparison of the algorithm in this paper with reachability tools for hybrid automata [...] a direct comparison for all four examples would be quite instructive.
>
> Thank you for the suggestion. The other examples could also not be solved by the generic approach. We agree that it would be interesting to provide further evidence, however (1) conceptually our approach is more efficient than a generic approach (explained above), and (2) since the generic approach could not provide any results, a tabular comparison is not possible.
>
> > given the amount of detail in Figures 4,5, and 6, it would help to make them higher resolution.
>
> Agreed.
>
> > the fixed point check seems to assume time-invariant dynamics. Is this correct? If so, why pass $t_0$ and $t_1$ as parameters to $post_f$ instead of simply passing $\tau$?
>
> Indeed, we consider time-invariant systems. Yes, we could pass tau instead most of the time, except potentially in the last time interval. The correct value is t1-t0. We can see that passing the time interval can lead to confusions. Since t1-t0 is unnecessarily complex, we will instead pass tau as suggested and assume that T is a multiple of tau to simplify the presentation.
>
> > The paper does not explicitly discuss the limitations and might benefit from a small discussion about the same.
>
> Our experiments already include a relatively large decision tree (depth 10, 177 nodes) for the drone quadrotor problem to show that large trees are also admissible, and as we argued in the paper, there exists a generic hybrid-system algorithm to verify DTCS, which is however impractical (Section 4.3). The main message of the paper is that a dedicated verification algorithm significantly outperforms such a generic algorithm.
>
> Nevertheless we agree that scalability is an interesting question, however in reachability analysis evaluating scalability is not straight-forward, since neither does 1) the size of the tree nor 2) the dynamics of the system impact the algorithm in a predictable way. Below we give examples for both points.
>
> 1) The size of the tree is not necessarily dominating because multiple leaves may share the same action, or some leaves may not be used at all during the execution. What matters is the number of times the decision boundaries are partially crossed by the reachable states after each time step. Given that different controllers may implement different policies, we cannot simply compare analyses with controllers of different size, since the verification performance may even improve for larger controllers. For instance, please see the two alternative controllers for the acrobot system (Fig. 9 in the supplementary material), whose behaviors differ significantly (Fig. 6). Note that in general we are interested in smaller trees since this is one of the main motivations for using decision-tree controllers over other (learned) non-interpretable controllers.
>
> 2) It is hard to characterize the 'size' of a dynamical system. For instance, the competition on verifying continuous and hybrid systems (see [1*]) showcases that most verification solvers struggle with the *three-dimensional* Robertson chemical reaction, whereas the *seven-dimensional* Laub-Loomis model has several solvers that easily solve it. Note that this part is orthogonal to our work since our contribution is not in the algorithm for dynamical systems.
>
> We will add this discussion to the paper.
>
> [1*] Geretti et al.: ARCH-COMP22 Category Report: Continuous and Hybrid Systems with Nonlinear Dynamics. 2022.

---

> > ### Comment · Reviewer_yJkM · 2023-08-15
> > **Response to rebuttal**
> >
> > Thank you for the detailed response to my questions. This is a nice piece of work and I will keep my score.

---

### Official Review · Reviewer_xs5Q · 2023-07-07

**Soundness:** 3 good
**Presentation:** 3 good
**Contribution:** 3 good
**Rating:** 6
**Confidence:** 3

**Summary:**

The paper puts forward a verification method for decision tree-based systems in
continuous time. The method implements a reachability algorithm that computes
over-approximations of the set of reachable states for a sequence of time
intervals until a time horizon is reached. The approximations at each step are
derived using Taylor models.

**Strengths:**

While the contribution is based on standard reachability-based verification
methods for hybrid and neural network-driven systems, the development of said
methods to verify decision trees is novel.

Good experimental evaluation which shows improvements over the state-of-the-art
in reachability for hybrid automata. This can in my view further improved  by
including the following:
    1. a discussion of the scalability of the method with respect to a wider
    range of model sizes.
    2. a discussion of the comparative amenability to reachability-based
    verification of decision tree-based and neural network-based systems.

**Weaknesses:**

The paper does not include a discussion on the scalability of the proposed
method.

**Questions:**

I have no questions. I thank the authors for the clarity of the text.

**Limitations:**

Adequately addressed.

---

> ### Author Rebuttal · Authors · 2023-08-09
>
> Thank you for the excellent review.
>
> > The paper does not include a discussion on the scalability of the proposed method.
>
> Our experiments already include a relatively large decision tree (depth 10, 177 nodes) for the drone quadrotor problem to show that large trees are also admissible, and as we argued in the paper, there exists a generic hybrid-system algorithm to verify DTCS, which is however impractical (Section 4.3). The main message of the paper is that a dedicated verification algorithm significantly outperforms such a generic algorithm.
>
> Nevertheless we agree that scalability is an interesting question, however in reachability analysis evaluating scalability is not straight-forward, since neither does 1) the size of the tree nor 2) the dynamics of the system impact the algorithm in a predictable way. Below we give examples for both points.
>
> 1) The size of the tree is not necessarily dominating because multiple leaves may share the same action, or some leaves may not be used at all during the execution. What matters is the number of times the decision boundaries are partially crossed by the reachable states after each time step. Given that different controllers may implement different policies, we cannot simply compare analyses with controllers of different size, since the verification performance may even improve for larger controllers. For instance, please see the two alternative controllers for the acrobot system (Fig. 9 in the supplementary material), whose behaviors differ significantly (Fig. 6). Note that in general we are interested in smaller trees since this is one of the main motivations for using decision-tree controllers over other (learned) non-interpretable controllers.
>
> 2) It is hard to characterize the 'size' of a dynamical system. For instance, the competition on verifying continuous and hybrid systems (see [1*]) showcases that most verification solvers struggle with the *three-dimensional* Robertson chemical reaction, whereas the *seven-dimensional* Laub-Loomis model has several solvers that easily solve it. Note that this part is orthogonal to our work since our contribution is not in the algorithm for dynamical systems.
>
> We will add this discussion to the paper.
>
> [1*] Geretti et al.: ARCH-COMP22 Category Report: Continuous and Hybrid Systems with Nonlinear Dynamics. 2022.

---

> > ### Comment · Reviewer_xs5Q · 2023-08-16
> > **Thank you for the response**
> >
> > Thank you for the detailed response which addresses my concern on scalability. My view of the paper remains positive.

---

### Official Review · Reviewer_jTr9 · 2023-07-07

**Soundness:** 3 good
**Presentation:** 3 good
**Contribution:** 3 good
**Rating:** 6
**Confidence:** 3

**Summary:**

This paper presents a method to solve the reach-avoid problem for dynamical systems controlled by a decision tree in continuous time. The authors assert that this is the first paper to solve the problem in the continuous time setting. In the paper, the authors first provide a good overview of decision trees, the use of decision trees as neural network surrogates, and verification of decision tree controlled systems (DTCS). They also explain why tools developed for verification of neural network controlled systems are not directly applicable to DTCS. The authors then explain their approach, which is based on a typical reachability algorithm and determines the set of reachable states for an iteration of the control loop.


**Strengths:**

1. The problem of DTCS verification is interesting and well motivated.
2. The paper is clearly and concisely written, and the justification of the paper as described is sound. The authors describe in great detail how their algorithms satisfy the equations necessary for the reach-avoid problem in DTCS. The authors also provide evaluations of their approach using multiple benchmark problems.

**Weaknesses:**

1. The novelty of the approach is not adequately explained. I believe that it is novel, but the authors need to explicitly state which parts of the algorithm are specifically designed for use with DTCS and which are just adapted from the typical reachability algorithm.
2. Section 3.2, which describes the generalization of the solution to unbounded time problems, could perhaps use more details.
3. The authors should also include some statement on how common axis-aligned predicates may be found in real-world DTCS. Is this a reasonable assumption?


**Questions:**

Please see the comments above on weakness.

**Limitations:**

The paper could use more discussion on its limitations, particularly regarding scalability. The work does not have obvious negative societal impact.

---

> ### Author Rebuttal · Authors · 2023-08-09
>
> Thank you for the excellent review.
>
> > The novelty of the approach is not adequately explained. I believe that it is novel, but the authors need to explicitly state which parts of the algorithm are specifically designed for use with DTCS and which are just adapted from the typical reachability algorithm.
>
> Algorithm 3 as well as Theorem 2 are completely new. We will restructure the paper to clarify this point. Compared to analyzing a DTCS with a generic hybrid-system tool, the most important aspects of the special structure our algorithm allows to exploit are as follows.
>
> (1) The decision tree partitions the set X. In particular, each predicate either cuts X in two or keeps it intact. Furthermore, if X satisfies a DT predicate, we do not have to explore the complement branch. In a general hybrid system, there is no "else case" and the conditions may overlap; hence the (expensive) intersection computations have to be performed repeatedly and the complement branch has to be analyzed as well.
>
> (2) If the set X only leads to leaves with the same action, we do not need to split it (this happens quite often), which avoids the main source for loss of precision altogether. In a general hybrid system, combined with aspect (1) above, the connection is lost.
>
> (3) Axis-aligned predicates allow to use (simple but efficient) box abstractions.
>
> > Section 3.2, which describes the generalization of the solution to unbounded time problems, could perhaps use more details.
>
> Thank you for the feedback. This section is indeed a very brief overview. The discussion is considered folklore in the reachability community, and the reason we included it is to be self-contained with respect to the experiments. In short, once the system finds itself in a state that was already analyzed previously, we can conclude that the system has reached a fixed point. This observation naturally generalizes to sets of states. We will make the description more formal.
>
> > The authors should also include some statement on how common axis-aligned predicates may be found in real-world DTCS. Is this a reasonable assumption?
>
> Axis-aligned predicates are very standard, also for DTCS. The tool Uppaal Stratego (David et al. 2015) learns such DT controllers, which are applied in industrial applications (e.g., to control smart homes [1*] and traffic lights [2*]). The tool dtControl (Ashok et al. 2020) also uses axis-aligned predicates.
>
> > The paper could use more discussion on its limitations, particularly regarding scalability.
>
> Our experiments already include a relatively large decision tree (depth 10, 177 nodes) for the drone quadrotor problem to show that large trees are also admissible, and as we argued in the paper, there exists a generic hybrid-system algorithm to verify DTCS, which is however impractical (Section 4.3). The main message of the paper is that a dedicated verification algorithm significantly outperforms such a generic algorithm.
>
> Nevertheless we agree that scalability is an interesting question, however in reachability analysis evaluating scalability is not straight-forward, since neither does 1) the size of the tree nor 2) the dynamics of the system impact the algorithm in a predictable way. Below we give examples for both points.
>
> 1) The size of the tree is not necessarily dominating because multiple leaves may share the same action, or some leaves may not be used at all during the execution. What matters is the number of times the decision boundaries are partially crossed by the reachable states after each time step. Given that different controllers may implement different policies, we cannot simply compare analyses with controllers of different size, since the verification performance may even improve for larger controllers. For instance, please see the two alternative controllers for the acrobot system (Fig. 9 in the supplementary material), whose behaviors differ significantly (Fig. 6). Note that in general we are interested in smaller trees since this is one of the main motivations for using decision-tree controllers over other (learned) non-interpretable controllers.
>
> 2) It is hard to characterize the 'size' of a dynamical system. For instance, the competition on verifying continuous and hybrid systems (see [3*]) showcases that most verification solvers struggle with the *three-dimensional* Robertson chemical reaction, whereas the *seven-dimensional* Laub-Loomis model has several solvers that easily solve it. Note that this part is orthogonal to our work since our contribution is not in the algorithm for dynamical systems.
>
> We will add this discussion to the paper.
>
> [1*] Larsen et al.: Online and Compositional Learning of Controllers with Application to Floor Heating. 2016.
> [2*] Bilgram et al.: Online and Proactive Vehicle Rerouting with Uppaal Stratego. 2020.
> [3*] Geretti et al.: ARCH-COMP22 Category Report: Continuous and Hybrid Systems with Nonlinear Dynamics. 2022.

---

> > ### Comment · Reviewer_jTr9 · 2023-08-17
> >
> > Thank you for addressing my comments. I am satisfied with the response and will raise my score accordingly.

---

### Decision · Program_Chairs · 2023-09-21

**Decision:**

Accept (spotlight)

**Comment:**

This paper proposes the reach-avoid problem for decision tree controlled systems in continuous time. The reach-avoid problem for decision tree controlled systems is to decide whether all trajectories reach the absorbing goal state without reaching the error set before the time limit. The goal is to overestimate the reachable states up to the time horizon using Taylor methods due to the undecidability of the problem.

The authors are encouraged to incorporated the feedback from the reviewers in the final version: (1) explicitly state which parts of the algorithm are specifically designed for use with DTCS and which are adapted from existing reachability/verification of dynamical systems algorithms, (2) improve the exposition on how the special structure of decision tree policies are exploited, (3) include the swing-up cart pole system results which the authors did not managed to do during the rebuttal phase, (4) improve the presentation quality of the whole paper, such as vectorization of all figures.

Overall, this is a nice contribution on verification of interpretable decision tree control systems to the community.